# Evolution of diversity in metabolic strategies

Rodrigo Caetano[1]*, Yaroslav Ispolatov[2], Michael Doebeli[3]

[1]Departamento de Física, Universidade Federal do Paraná, Curitiba, Brazil;
[2]Department of Physics, University of Santiago of Chile (USACH), Santiago, Chile;
[3]Department of Mathematics and Department of Zoology, University of British Columbia, Vancouver, Canada

**Abstract** Understanding the origin and maintenance of biodiversity is a fundamental problem. Many theoretical approaches have been investigating ecological interactions, such as competition, as potential drivers of diversification. Classical consumer-resource models predict that the number of coexisting species should not exceed the number of distinct resources, a phenomenon known as the competitive exclusion principle. It has recently been argued that including physiological tradeoffs in consumer-resource models can lead to violations of this principle and to ecological coexistence of very high numbers of species. Here, we show that these results crucially depend on the functional form of the tradeoff. We investigate the evolutionary dynamics of resource use constrained by tradeoffs and show that if the tradeoffs are non-linear, the system either does not diversify or diversifies into a number of coexisting species that do not exceed the number of resources. In particular, very high diversity can only be observed for linear tradeoffs.

## Introduction

Life on Earth is spectacularly diverse (*May, 1988*). For example, one study in the early 2000s found that the number of species of fungi is, by a conservative estimate, ca. 1.5 million (*Hawksworth, 2001*), which was subsequently revised to be between 2.2 and 3.8 million species (*Hawksworth and Lücking, 2017*). Microbes are by far the most diverse form of life. They constitute approximately 70–90% of all species (*Larsen et al., 2017*). Perhaps even more astonishing than the number of species is the fact that all of them came from a single common ancestor (*Darwin, 1859*; *Steel and Penny, 2010*; *Theobald, 2010*). To understand the fundamental mechanisms behind such diversification is one of the most relevant problems addressed by the scientific community (*Mayr and Mayr, 1963*; *Coyne, 1992*; *Rice and Hostert, 1993*; *Higashi et al., 1999*; *Dieckmann and Doebeli, 1999*; *Gavrilets and Waxman, 2002*; *de Aguiar et al., 2009*; *Doebeli, 2011*).

Recently, ecological interactions, such as competition, have received a lot of attention as potentially very strong drivers of diversification and speciation. A widely used class of models in which this phenomenon can be observed is based on classical Lotka-Volterra competition models, which are augmented by assuming that the carrying capacity is a (typically unimodal) function of a continuous phenotype, and that the strength of competition between two phenotypes is measured by a competition kernel, which is typically assumed to be a (symmetric) function of the distance between the competing phenotypes, with a maximum at distance 0 (so that the strength of competition decreases with increasing phenotypic distance).

These assumptions are biologically plausible, and such models have been widely used to provide insights into evolutionary diversification due to competition (*Dieckmann and Doebeli, 1999*; *Doebeli and Ispolatov, 2010*; *Doebeli and Ispolatov, 2017*). However, these models are not derived mechanistically from underlying resource dynamics, and in fact it is known that the

**\*For correspondence:**
caetano@fisica.ufpr.br

commonly used Gaussian functions for the carrying capacity and the competition kernel are not compatible with resource-consumer models (*Abrams, 1986*; *Ackermann and Doebeli, 2004*). A more mechanistic approach is desirable.

Recently, a MacArthur consumer-resource model (*Macarthur and Levins, 1967*) was studied in an ecological context with a view toward explaining the existence of very high levels of diversity (*Posfai et al., 2017*; *Erez et al., 2020*). The authors consider different species competing for $p$ interchangeable resources, each supplied at a constant rate (*Posfai et al., 2017*) or periodically repleted after being used (*Erez et al., 2020*). A consumer species is characterized by an uptake strategy, $\alpha = (\alpha_1, ..., \alpha_p)$, where the $j$ th component $\alpha_j \geq 0$ represents the amount of cellular metabolism allocated to the uptake of the $j$ th resource. The rate of consumption of the $j$ th resource and thus its contribution to the growth rate is assumed to be proportional to $\alpha_j$. The total amount of cellular metabolism available for resource uptake is limited, and hence it is natural to assume a tradeoff between the uptake rates of different resources. In general mathematical terms, a tradeoff is typically given by a function $T(\alpha) = T(\alpha_1, ..., \alpha_p)$ that is increasing in each of the arguments $\alpha_j$, and such that the only permissible allocation strategies $\alpha$ are those satisfying $T(\alpha) \leq E$, where $E$ is a constant. The analysis is then typically restricted to the subspace of strategies defined by $T(\alpha) = E$ (because $T$ is increasing in each $\alpha_j$). It was shown in *Posfai et al., 2017*; *Erez et al., 2020* that, under the assumption of a linear tradeoff, $\sum_j^p \alpha_j = E$, very high levels of diversity, that is, many different species with different $\alpha$-strategies, can coexist. This is a very interesting finding because it violates the competitive exclusion principle (*Hardin, 1960*), according to which at most $p$ different species should be able to stably coexist on $p$ different resources. Such high levels of diversity emerging from simple consumer-resource models could help solve the paradox of the plankton (*Hutchinson, 1961*) from an ecological perspective.

However, metabolic tradeoffs are not necessarily linear, and in fact there is reason to believe that they almost never are. Nature owes its complexity and diversity to the non-linearity of the underlying physical and chemical processes. In particular, the non-linearity of tradeoffs is an essentially inevitable consequence of the general non-linearity of chemical kinetics. The rate and mass action equilibrium of even a simple bimolecular reaction are in general non-linear functions of the concentrations of reactants. Linear approximations are commonly used when the concentrations of certain reactants are vastly exceeding the concentrations of others, or when the binding is so strong that the dissociation constant of a complex is much less than typical concentrations of its constituents. However, while the concentrations of enzymes in bacteria (which are probably the most realistic prototype for models of *Posfai et al., 2017*; *Erez et al., 2020*) are generally below those of their substrates, the difference is often only few- or 10-fold, which is insufficient to approximate the enzymatic kinetics by functions that are linear in enzymatic concentrations. For example, a detailed study (*Bennett et al., 2009*) of the model microbe *Escherichia coli* revealed that out of 103 metabolites, 35 have concentrations above 1 mM, but the concentrations of 46 metabolites are in tens or single micromole digits, including two metabolites with concentrations below 1 µM. Supporting this, BIONUMBERS (*Milo et al., 2010*) estimate the typical metabolite concentration in an *E. coli* bacterium as 32 µM. At the same time, BIONUMBERS provide the evidence for concentrations of important *E. coli* glycolysis enzymes in tens and even hundreds of µM, and hence the difference between metabolite and enzyme concentrations generally does not seem to be large enough to justify linear approximations.

Another argument for the prevalence of non-linearity in tradeoffs is based on the oligomerization of more than half of all metabolic enzymes (*Marianayagam et al., 2004*). The dissociation constants of dimer or oligomer enzymes is often comparable to the concentrations of its monomer units to make the dimerization sensitive to environmental conditions and use it as a regulator of enzymatic activity (*Ali and Imperiali, 2005 Traut, 1994*). Thus, doubling the concentration of an oligomer requires more (in case of hetero-oligomer) or less (in case of homo-oligomer) than doubling the concentrations of its monomers, and hence the metabolic costs of the former in terms of the metabolic costs of the latter are non-linear.

Since metabolic tradeoffs can often be expected to be non-linear, here we generalize the models of *Posfai et al., 2017*; *Erez et al., 2020* by incorporating non-linear tradeoffs in resource use. Specifically, we consider energy budgets of the form

$$\sum_{j=1}^{p} \alpha_j^{\gamma} = E, \tag{1}$$

where $\gamma$ and $E$ are positive constants.

In addition, we incorporate evolutionary dynamics into the ecological models of *Posfai et al., 2017*; *Erez et al., 2020*, which allows us to investigate not only the conditions under which diversity can be maintained, but also the evolution of diversity from a single ancestral species. We show that in the resulting evolutionary model, coexistence of more than $p$ species only emerges for the (structurally unstable) linear case $\gamma = 1$. Using adaptive dynamics and numerical simulations, we show that regardless of the value of $\gamma$, an initially monomorphic population always evolves to an attractive fixed point (also called 'singular point'), after which two generic scenarios are possible: (i) if $\gamma < 1$, the population branches and diversifies, with the maximal number of coexisting species equal to the number of resources $p$, a state in which each species is a complete specialist on exactly one of the resources; (ii) if $\gamma > 1$, an initially monomorphic population also evolves to a singular point, but subsequently does not diversify and instead remains a monomorphic generalist.

To make the argument for the relevance of non-linear tradeoffs even more solid, we prove that an omnipresent non-linearity in the dependence of nutrient uptake rates on $\alpha$ can be transformed into the non-linearity of tradeoff (*Equation 1*), and vice versa. Thus, a non-linearity in either the tradeoff or the metabolic rates is sufficient to bring the diversity down to the competitive exclusion limit. We also show that the two scenarios (of either a generalist or $p$ specialists) emerge as a result of purely ecological dynamics in a system initially populated with multiple species with different uptake strategies $\alpha$ that satisfy (*Equation 1*).

Overall, our results show that very high levels of diversity do not evolve in the consumer-resource model considered here in a realistic scenario where tradeoffs in resource preference or the resource uptake rates are non-linear.

## Model and results

We consider a population competing for $p$ substitutable resources in well-mixed environments. A phenotypic species $\alpha$ is characterized by its metabolic allocation strategy $\alpha = (\alpha_1, \ldots, \alpha_p)$, where $\alpha_j$ is the per capita rate at which individuals of species $\alpha$ take up the $j$ th nutrient. Various coexisting species are distinguished by their specific $\alpha$'s. From a physiological perspective, $\alpha_j$ is proportional to the amount of metabolic effort allocated by the individuals of species $\alpha$ to capture nutrient $j$. Intrinsic limitations on metabolic activities impose a restriction on the total amount of nutrient uptake. For simplicity, we assume that this intrinsic limitation leads to a tradeoff in the components $\alpha_j$ of the form (*Equation 1*). (Note that we also assume $\alpha_j \geq 0$ for all $j$.) Throughout, we will set the scaling parameter $E = 1$. (See Appendix 1 for a more general treatment, in which the exponent $\gamma$ can differ for different directions $\alpha_j$ in phenotype space.)

Following *Posfai et al., 2017*, we denote by $c_j(t)$ the concentration of resource $j$ at time $t$, and we assume that the amount of resource $j$ available for uptake per individual (e.g., the amount of resource bound to the outer membrane of a microbial cell) is given by a monotonously increasing function $r_j(c_j)$. Specifically, we assume this function to be of Monod type, $r_j(c_j) = c_j/(K_j + c_j)$. Thus, the rate of uptake of resource $j$ by an individual consumer with uptake strategy $\alpha$ is $\alpha_j r_j(c_j)$.

## Chemostat conditions

We assume that resources are supplied to the system at a constant rate defined by the supply vector $s = (s_1, \ldots, s_p)$, so that resource $j$ is supplied at a constant total rate $s_j$ and decays at a rate $\mu_j$ (*Posfai et al., 2017*). This generates the following system of equations for the ecological dynamics of the concentrations $c_j$, $j = 1, \ldots, p$:

$$\frac{dc_j}{dt} = s_j - \left(\sum_{\alpha} n_{\alpha}(t)\alpha_j\right) r_j(c_j) - \mu_j c_j. \tag{2}$$

Here, $n_{\alpha}(t)$ is the population density of species $\alpha$ at time $t$, so that $\sum_{\alpha} n_{\alpha}(t)\alpha_j$ is the total amount of metabolic activity invested into uptake of resource $j$ (the sum runs over all species $\alpha$ present in the community). We further assume that the cellular per capita birth rate of species $\alpha$ is equal to the

amount of nutrient absorbed by each individual. The dynamics of the population density $n_\alpha$ then becomes

$$\frac{dn_\alpha}{dt} = \left( \sum_{j=1}^{p} \alpha_j r_j(c_j) - \delta \right) n_\alpha,$$
(3)

where $\delta$ is the per capita death rate, which is assumed to be the same for all consumers.

The evolutionary dynamics of the the traits $\alpha_j$ can be solved analytically only for a simplified system in which the resource decay (dilution) rates $\mu_j$ are set to 0. This assumption, also made in *Posfai et al., 2017*, corresponds to rapid consumption of almost all resource. In Appendix 1, we derive the adaptive dynamics for the allocation strategies, that is, for the traits $\alpha_j$ (*Metz et al., 1992*; *Dieckmann and Law, 1996*; *Dieckmann and Doebeli, 1999*; ; *Hui et al., 2018*; *Doebeli, 2011*; *Geritz et al., 1997*). We show that with vanishing decay rates, there is a unique singular point

$$\alpha_j^* = \left( \frac{s_j}{\sum_{k=1}^{p} s_k} \right)^{\frac{1}{\gamma}}.$$
(4)

Calculations of the Jacobian of the adaptive dynamics (an indicator of convergence stability of a fixed point) and of the Hessian of the invasion fitness function (which distinguishes whether the fixed point is an evolutionary endpoint or a branching point) yield the following conclusions: Regardless of the value of $\gamma$, the singular point $\alpha^*$ is always convergent stable, so that the system approaches $\alpha^*$ from any initial condition. If $\gamma > 1$, the singular point $\alpha^*$ is also evolutionarily stable and hence represents the evolutionary endpoint. In particular, no diversification takes place. On the other hand, if $\gamma < 1$, the singular point is evolutionarily unstable and hence is an evolutionary branching point. In particular, if $\gamma < 1$, the system will diversify into a number of coexisting consumer species. If $\gamma = 1$ (linear tradeoff), the fitness Hessian is 0, representing evolutionary neutrality.

To check our analytical approximations and to investigate the details of diversification after convergence to the evolutionary branching point, we performed numerical simulations of evolving populations consisting of multiple phenotypic strains. The simulations were performed without the simplifying assumption of zero resource degradation (dilution) rates; further details of the numerical simulations are presented in Appendix 1.

In the figures below we show evolving populations as circles with radii proportional to the square root of population size $n_\alpha$ in three-dimensional strategy space $(\alpha_1, \alpha_2, \alpha_3)$, viewed orthogonally to the simplex plane $\sum_{i=1}^{3} \alpha_i = 1$. With the constraint $\sum_{i=1}^{3} \alpha_i^\gamma = 1$, the coordinates of each population are $(\alpha_1^\gamma, \alpha_2^\gamma, \alpha_3^\gamma)$. In the following numerical examples, we considered a symmetric supply of resources $s_i = 1$ and a slow resource degradation, $\mu_i K_i = 0.1$.

We first consider scenarios with linear tradeoffs, $\gamma = 1$. *Figure 1* shows the evolution of a population (shown in blue circles) whose individuals die at constant rate $\delta = 1$ (corresponding videos of the simulations can be accessed through the links provided in the figure legends). The black circle represents the singular point that is calculated in the limit of low degradation of nutrients, given by *Equation 4*. *Figure 1(a)* shows the initial monomorphic population far from the singular point. An intermediate time of the evolutionary process is shown in *Figure 1(b)*, in which the population remains monomorphic and is approaching the singular point $\alpha^*$. For $\gamma = 1$, the singular point is neutral evolutionarily (all eigenvalues of the Hessian of the invasion fitness function are 0 due to the linearity of the tradeoff), and once the population converges to the singular point, it starts to diversify 'diffusively', as anticipated in *Posfai et al., 2017*: neutrality of selection results in communities consisting of a large number of species. Thus, the high diversity observed in this case is an evolutionary consequence of the selective neutrality caused by a linear enzymatic tradeoff.

The situation changes for non-linear tradeoffs, $\gamma \neq 1$, which generates two very different evolutionary regimes depending on whether $\gamma > 1$ or $\gamma < 1$ (even when the deviation of $\gamma$ from one is small). *Figure 2(a–c)* shows an example of the evolutionary dynamics for $\gamma = 1.1$.

The dynamics starts with an initial monomorphic population far from the singular point, as shown in *Figure 2(a)*. As in the linear case, and as predicted by the analytical theory, the monomorphic population converges toward the singular point *Figure 2(b)*. However, because $\gamma > 1$ the singular point is evolutionarily stable, and no diversification occurs (apart from mutation-selection balance around the singular point). Instead, when the population reaches the singular point, evolution comes

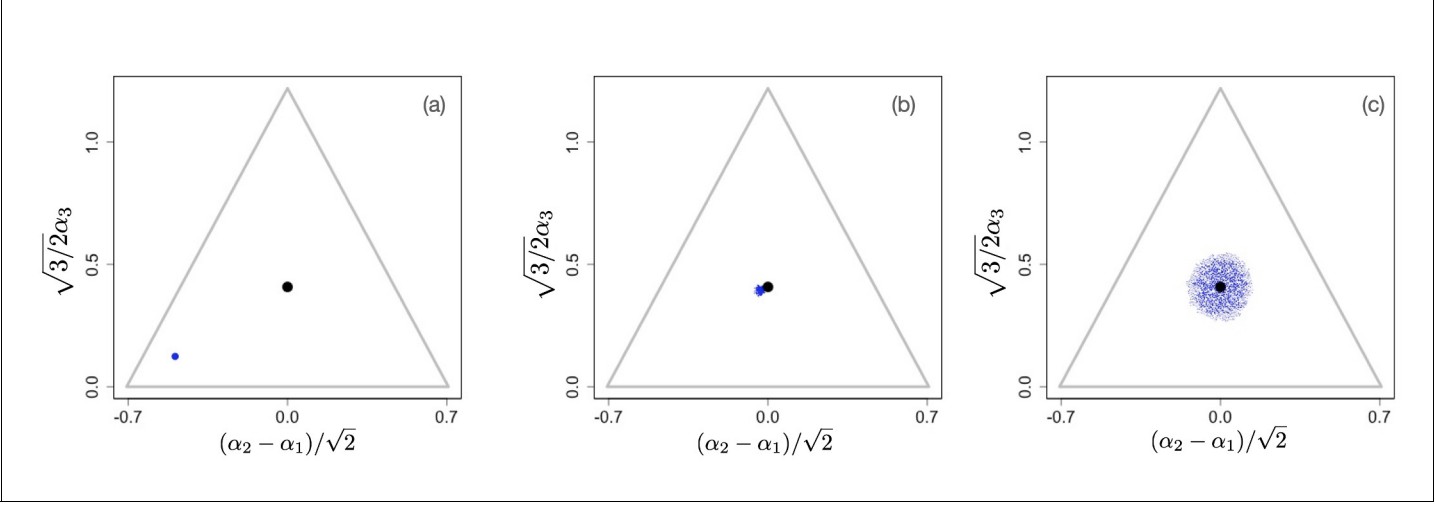

**Figure 1.** Snapshots illustrating the beginning, intermediate, and advanced stages of evolution under a linear constraint, $\gamma = 1$ . A video of the entire evolutionary process can be found here, frames are recorded every 200 time units until $t$=30,000 and then, to better illustrate slow neutral evolution, the frame recording times $t_i$ were defined as a geometric progression $t_{i+1} = 1.006t_i$. Other parameter values were $s_j = 1$, $\mu_j K_j = 0.1$ for $j = 1, 2, 3$, and $\delta = 1$.

to a halt, and all individuals are generalists, that is, use all resources to some extent (as determined by the location of the singular point), as depicted in *Figure 2(c)*.

On the other hand, *Figure 3(a–c)* shows the evolutionary process for a community with $\gamma = 0.9$. The initial configuration is shown in *Figure 3(a)*. As in the previous examples, the initial phase of evolution ends with the population converging to the singular point $\alpha^*$. However, in this case, the singular point is an evolutionary branching point giving rise to the emergence of distinct and diverging phenotypic clusters (*Figure 3(b)*). The final state of the evolutionary process is shown in *Figure 3(c)*: there are three coexisting phenotypic clusters, each being a specialist in exactly one of the resources. Our numerical simulations indicate that the results shown in *Figures 1–3* are general and robust: non-neutral diversification occurs only for $\gamma < 1$ and typically leads to coexistence of $p$ specialists. In fact, the results easily generalize to situations in which the exponent $\gamma$ in the tradeoff function may be different for different directions in phenotype space, that is, for different $\alpha_j$. As we show in Appendix 1, evolutionary branching along a direction $\alpha_j$ in phenotype space can occur if the

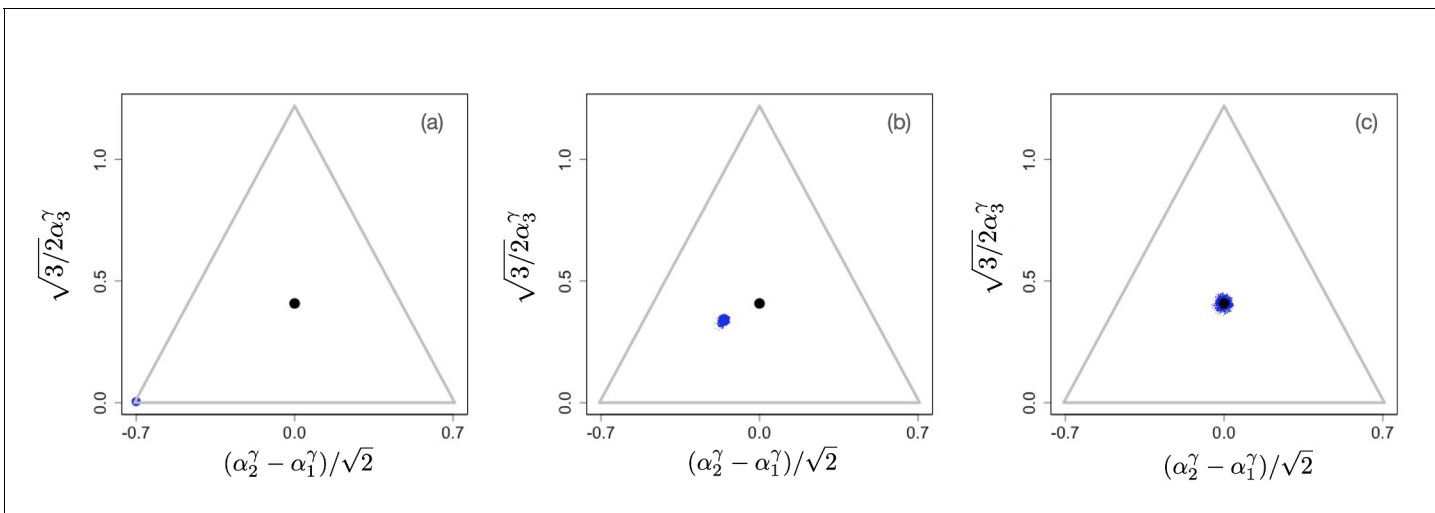

**Figure 2.** Example of evolutionary dynamics for $\gamma = 1.1$, showing convergence to the singular point given by *Equation 4* (and indicated by the black dot), but no subsequent diversification. The corresponding video can be found here , each frame in the video is separated by 1,000 time steps. Other parameter values were $s_j = 1$, $\mu_j K_j = 0.1$ for $j = 1, 2, 3$, and $\delta = 0.25$.

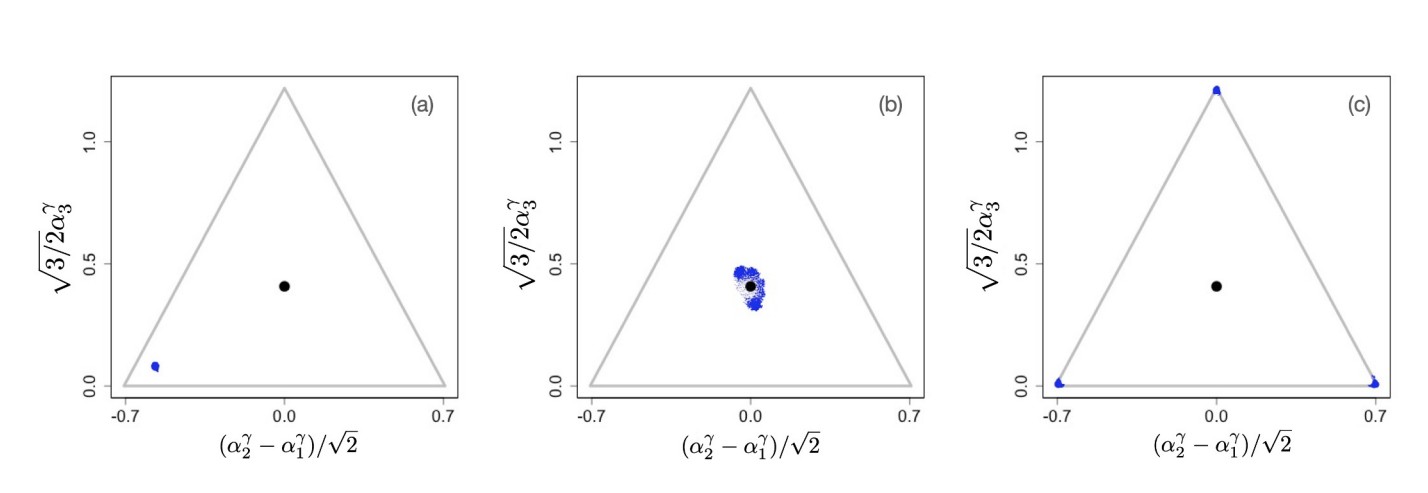

**Figure 3.** Example of evolutionary dynamics for $\gamma = 0.9$, showing initial convergence to the singular point (indicated by the black dot) and subsequent diversification into three specialists, each consuming exclusively one of the three resources. The corresponding video can be found here, each frame in the video is separated by 1,000 time steps. Other parameter values were $s_j = 1$, $\mu_j K_j = 0.1$ for $j = 1, 2, 3$, and $\delta = 0.25$.

corresponding exponent $\gamma_j < 1$. *Appendix 1—figure 2* and *Appendix 1—figure 3* in Appendix 1 illustrate scenarios in which only a subset of the phenotypic directions $\alpha_j$ are branching directions along which evolutionary diversification occurs. In such a case, the number of distinct species resulting from the evolutionary process is less than $p$.

Finally, we note that our results for the effects of non-linear tradeoffs on evolutionary dynamics have corresponding results in purely ecological scenarios, such as those studied in *Posfai et al., 2017*. We simulated ecological time scales by seeding the system with a set of for example randomly chosen phenotypes throughout phenotype space and running the population dynamics with the mutational process turned off. Again, as shown in *Appendix 1—figure 4*, non-linear tradeoffs have a profound effect on the number of surviving species in such ecological simulations, with many species coexisting when $\gamma = 1$, as reported in *Posfai et al., 2017*, but with typically only $p$ species surviving when $\gamma < 1$ and only very few species surviving in the close vicinity of the singular point when $\gamma > 1$.

## Serial dilution conditions

Serial dilution conditions are defined as a sequence of explicitly non-stationary inoculation and growth events (*Erez et al., 2020*), which mimics seasonality or batch culture experiments (e.g., *Lenski and Travisano, 1994*). Each growth phase starts with the introduction of a diluted collection of species from a previous batch

$$n_\alpha(0) = \rho_0 \frac{n_\alpha(t_{fin})}{\sum_\alpha, n_\alpha, (t_{fin})}, \tag{5}$$

into a fresh batch of resources with a given composition $c_j(0)$. In each batch, the species densities $n_\alpha(0)$ increase with time as

$$\frac{dn_\alpha}{dt} = \left( \sum_{j=1}^{p} \alpha_j r_j(c_j(t)) \right) n_\alpha, \tag{6}$$

while resources are depleted:

$$\frac{dc_j}{dt} = - \left( \sum_\alpha n_\alpha(t) \alpha_j \right) r_j(c_j). \tag{7}$$

Unlike in the chemostat model, the death of individuals and the decay of resources are ignored ($\delta = 0$ and $\mu = 0$). Each event ends at time $t_{fin}$ when all resources are almost completely depleted,

$$\sum_{j=1}^{p} c_j(t_{fin}) = c_{fin} \approx 0, \tag{8}$$

and the process is repeated.

Due to the explicit non-stationarity of such serial dilution processes, one of the main assumptions of our adaptive dynamics analysis, the stationarity of resident populations, is not satisfied. Nevertheless, our numerical simulations show that the conclusions drawn for the chemostat case also hold for the serial dilution conditions, to the point that the simulation snapshots are visually indistinguishable from those shown in *Figure 2* and *Figure 3*. However, in the videos, which can be found here , it is possible to see the oscillating population density, caused by the serial dilution protocol.

Specifically, we simulated the serial dilution for three limits considered in *Erez et al., 2020*, $c_j(0) = 10K$, $c_j(0) = K$, and $c_j(0) = 0.1K$ for $\rho_0 = 10^{-3}$ and $c_{fin} = 10^{-8}$. All other parameters were the same as used in *Figure 1*, *Figure 2*, *Figure 3* and corresponding videos.

In all three cases $c_j(0) \gg K$, $c_j(0) \sim K$, and $c_j(0) \ll K$, we observed that for $\gamma > 1$, the monomorphic population converges toward the singular point $\alpha^*$ (*Figure 2(b)*) and video files here . The singular point is evolutionarily stable, hence, as shown in *Figure 2(c)*, no subsequent diversification occurs (apart from narrow mutation-selection spreading around the singular point).

On the contrary, *Figure 3(a–c)* and videos accessible here show the evolutionary process for a community with $\gamma < 1$. The initial configuration is shown in *Figure 3(a)*. As in the previous examples, in the initial phase the monomorphic population evolves close to the singular point $\alpha^*$. However, in this case, the singular point is again an evolutionary branching point giving rise to the emergence of distinct and diverging phenotypic clusters (*Figure 3(b)*). The final state of the evolutionary process is shown in *Figure 3(c)*: there are three coexisting phenotypic clusters, each being a specialist on one of the resources.

In addition, purely ecological (i.e., mutationless) simulations performed similarly to what is described above and in *Erez et al., 2020* resulted in similar outcomes as in the chemostat model. In a system initially filled with many (200) species, only a few species survive after a fairly short transitory time. When $\gamma > 1$, one or a few species remain very close to the singular point $\alpha^*$, while for $\gamma < 1$, typically $p$ specialist species remain in the system. The videos of pure ecological simulations can be seen here.

Once evolution has come to its steady state, resulting in a single generalist species when $\gamma > 1$ or $p$ specialist species when $\gamma < 1$, each species is represented by a 'cloud' of phenotypes $\alpha$ (Panel C in *Figure 2* and *Figure 3*). Such a cloud is formed by a competition between the deterministic selection gradient that acts toward the center of the cloud, making the survival of peripheral species less likely, and the stochastic mutational process that broadens the distribution of strains in all directions, 'reseeding' new strains everywhere in the cloud, including its periphery. This is analogous to the classical mutation-selection balance occurring with stabilizing selection. When stabilizing selection is relatively weak, which occurs when the tradeoff is only weakly non-linear (with $\gamma$ close to 1), the dispersion of phenotypes around the centers of clouds is larger. Technically, this can be concluded from the factor $1 - \gamma$ in *Equation A20*. Thus, we make a potentially testable predictions that a weaker non-linearity in tradeoffs or uptake rates should result in broader distributions of corresponding phenotypes within specialist or generalist species. At the same time, the perspective of mutation-selection balance makes it easier to see the difference between the neutral evolutionary scenario of linear tradeoffs and the weakly non-linear case: While in the former case the distribution of strains will be uniform across the simplex (constrained only by the 'convex envelope' condition; *Posfai et al., 2017*), the non-linear tradeoffs lead to distinct species with well-localized distributions of phenotypes for any $\gamma \neq 1$. In an analogy with critical phenomena in physics, correlations typically decay exponentially, except at critical points, where they are exceptionally long-ranged. Such an 'anomalous' behavior requires careful tuning of parameters to get exactly to the critical point, unless the system is 'self-organized critical'. A similar situation appears to be the case with linear tradeoffs. It could be possible in principle that a system possesses carefully adjusted metabolic parameters so that for a range of uptake rates, the tradeoffs in enzyme concentrations are linear. Yet there appears

to be no evolutionary reason for 'self-organization' to such a state, and the accidental cancelation of all non-linearities is very unlikely.

## Discussion

To understand the origin and maintenance of diversity is a fundamental question in science. In particular, the mechanisms of diversification due to ecological interactions still generate lively debates.

Recently, tradeoffs in the rates of uptake of different resources were suggested as a mechanism to generate large amounts of diversity (*Posfai et al., 2017*; *Erez et al., 2020*), possibly solving the 'paradox of the plankton' (*Hutchinson, 1961*), and violating the competitive exclusion principle (*Hardin, 1960*), which states that the number of coexisting species should not exceed the number of resources. It has been shown that enzymatic allocation strategies that are plastic instead of fixed, so that individuals can change their allocation (while maintaining a linear tradeoff under a fixed allocation budget) in response to resource availability during their lifetime, tend to reduce the amount of diversity maintained in the ecological communities (*Pacciani-Mori et al., 2020*). Perhaps this is not surprising, since more plastic strategies tend to be able to be more generalist as well. As in *Posfai et al., 2017*; *Erez et al., 2020*, here, we consider the case of non-plastic strategies, in which each individual is defined by its allocation vector $\alpha$, but assuming a more general, non-linear form of tradeoffs. Moreover, we investigate evolutionary rather than just ecological dynamics to determine the conditions under which evolutionary diversification can occur. There are no true jacks-of-all trades in biology and tradeoffs are a ubiquitous assumption in evolutionary thinking and modeling. However, the cellular and physiological mechanisms that underly such tradeoffs are typically very complicated and the result of biochemical interactions between many different metabolic pathways. Attempts have been made to understand tradeoffs more mechanistically, particularly in microbes (*Litchman et al., 2015*), but higher-level modeling efforts most often still require a mostly phenomenological approach to incorporating tradeoffs. In this paper we assumed that each of $p$ resources is available to each microbial organism at a certain rate that depends on the resource concentration in the system. The microbe in turn is described phenotypically by the metabolic allocation strategy that defines its uptake of the available resources.

Without tradeoffs, and everything else being equal, the best strategy would be to allocate an infinite amount (or at least the maximal amount possible) of metabolic activity to every resource, a scenario that is generally unrealistic biologically. Rather, tradeoffs inherent to cell metabolism prevent such strategies. Formally, tradeoffs are given by one or more equations (or more generally inequalities) that the phenotypes of individuals have to satisfy.

In our simplistic models, tradeoffs are determined by the parameter $\gamma$, which essentially describes the curvature of the tradeoff function, with the linear tradeoff $\gamma = 1$ being the threshold between concave ($\gamma<1$) and convex ($\gamma>1$) tradeoffs. Formally, linear tradeoffs are the simplest case, but there is no a priori general reason why tradeoffs should be linear. Our results show that generically, diversity only evolves with concave tradeoffs, and the number of coexisting species never exceeds the number of resources. Only in the structurally unstable linear case ($\gamma = 1$), it is possible for very high levels of diversity to evolve due to the cessation of selection at the evolutionary equilibrium. Any value of $\gamma \neq 1$ precludes high amounts of diversity. Extensive numerical explorations revealed that these results are robust and qualitatively independent of particular parameter choices, such as the number of resources or the dynamics of resource input.

Furthermore, in Appendix 1 we show that the originally non-linear tradeoffs can be made linear by re-defining uptake rates $\alpha_i$ (*Equation A8*), thus 'transferring' the non-linearity to the nutrient uptake and the birth rate functions (*Equation A8*). But a metabolic and nutrient uptake rate is itself a linear function in the enzyme concentration only when the concentration of the substrate vastly exceeds the enzyme concentration. A good example is the well-known Michaelis-Menten approximation, which is identical to the formula used in *Posfai et al., 2017*; *Erez et al., 2020* for the dependence of nutrient uptake on enzyme allocation $\alpha$. While such linear approximations have been successfully applied in chemical kinetics for over a century, often without questioning their formal validity, the effect of linearization on ecological and evolutionary properties turns out to be very significant. The Michaelis-Menten kinetics is valid when the formation of enzyme-substrate complexes does not reduce the concentration of free substrate. Yet the intracellular concentration of enzymes in bacteria are often comparable to or are just few- or 10-fold smaller than those of their substrates

(*Bennett et al., 2009*; *Milo et al., 2010*). In Appendix 1 we sketch a derivation of kinetics of an enzymatic reaction in the general case assuming the steadiness of the concentration of the enzyme-substrate complex, but without the assumption that the enzyme concentration is negligible compared to that of the substrate. It follows that enzymatic reaction rates are generally sublinear in the concentrations of enzymes, which is intuitively clear from considering the rate saturation in the limit of infinite enzyme concentrations. However, sublinear rates are not the only possible deviation from linearity: the formation of enzyme oligomers (*Marianayagam et al., 2004*, *Traut, 1994*; *Ali and Imperiali, 2005*; *Traut, 1994*) and spatially organized complexes (*Schmitt and An, 2017*) are controlled by intrinsically non-linear (superlinear in case of homo-oligomers) mass action equilibria, thus making the enzymatic rates generally sigmoid functions (*Ricard and Noat, 1986*) of the amount of enzyme. Again, it follows that the physiological costs of the production of individual enzymes are typically non-linear.

There are also more direct ways to demonstrate the ubiquity of non-linear dependences of metabolic rates or fluxes $f$ on enzyme concentrations $\alpha$, for which quantities known as reaction elasticities or flux control coefficients are normally defined as double-logarithmic derivative, $d\ln[(f(\alpha))]/d\ln(\alpha)$. For example, for a general power law $f(\alpha) \equiv C\alpha^{\gamma}$ that we used to define metabolic tradeoffs (or uptake rates, see Appendix 1), the log-log derivative is equal to $\gamma$, the non-linearity parameter. For the tradeoffs used in *Posfai et al., 2017*; *Erez et al., 2020*, this derivate is always 1. However, it is not surprising that realistic assessments of such coefficients (e.g. *Loder et al., 2016*; *Giersch, 1995*; *Sun and Qian, 2002*; *Saavedra et al., 2005*; *Rohwer et al., 2000*; *Rutkis et al., 2013*; *Schmidt et al., 2016*; *van der Vlag et al., 1995*; and many other references) produce values that rarely come close to 1, and hence that the measured dependencies of metabolic fluxes on enzyme concentrations are significantly non-linear. For an easier parametrization of these non-linearities, it was suggested to express rates of complex enzymatic reactions as products of power-law functions of concentrations of enzymes and substrates (*Savageau, 1969*). This idea, originally suggested more than 50 years ago, has since developed a substantial following, which once again indicates the necessity to account for non-linearity in the kinetics of enzymatic pathways. All this indicates that reaction elasticities and flux control coefficients are typically distinct from one, which is essentially the main *raison d'être* for those quantities and for the science of metabolic engineering itself.

Whether sub- or super-linear, any deviation of the growth rates from the linear form (*Equation 3*) and *Equation A3* results in a revalidation of the competitive exclusion limit, similarly to non-linearity in tradeoffs. This serves as another indication that linear tradeoffs in metabolic rates is a biologically unrealistic and exceptional case, while generic non-linearities do not generate high levels of diversity, and instead the outcomes are in line with classical results about the evolution of resource generalists vs. resource specialists (*Ma and Levin, 2006*).

It is well known that the shape of tradeoff curves is, in general, an important component in adaptive dynamics models (*Kisdi, 2006*; *Kisdi, 2015*). In particular, studies of evolution of cooperation (e.g. *Damore and Gore, 2012*; *Archetti and Scheuring, 2012*) have stressed that the outcome of evolution is conditional on the curvature of the public good and cost functions and provided numerous biochemical reasons for non-linearity of metabolic rates in enzyme concentrations. Here, we have shown the importance of the tradeoff curvature for the evolution and maintenance of diversity in a general consumer-resource model. Of course, many potentially important ingredients that could yet lead to high or low diversity in these models were not considered in the present work. For example, dynamic and optimal metabolic strategies (*Pacciani-Mori et al., 2020*) and cross-feeding have recently been suggested as factors that could potentially enable such diversity (*Goyal and Maslov, 2018*), while 'soft constraints' that allow random deviations of metabolic strategies from the exact tradeoff constraint were reported in *Cui et al., 2020* to reduce the diversity even below the competitive exclusion limit. It will be interesting to consider these model extensions with non-linear tradeoffs.

Furthermore, it is possible that non-equilibrium ecological dynamics can allow for the maintenance of excess diversity. While this is not the case for externally imposed batch culture dynamics, as reported in the present paper, we have recently shown, using a different ecological model (*Doebeli et al., 2021*), that endogenous non-stationary 'boom-bust' population dynamics can lead to a significant increase in diversity above the saturation limit expected with equilibrium population dynamics. Together with many experimental results reporting non-stationarity and apparent chaoticity of the population dynamics of actual plankton species, this leads to the conjecture that rather

than the neutral evolutionary regime predicted in *Posfai et al., 2017*, non-stationary population dynamics induced by competition and predation (and perhaps external factors) may be more important in explaining high levels of diversity in natural systems.

## Additional information

### Competing interests
Michael Doebeli: Reviewing Editor, *eLife*. The other authors declare that no competing interests exist.

### Funding

| Funder | Grant reference number | Author |
|---|---|---|
| FONDECYT | 1200708 | Yaroslav Ispolatov |
| NSERC | Discovery Grant 219930 | Michael Doebeli |

The funders had no role in study design, data collection and interpretation, or the decision to submit the work for publication.

### Author contributions
Rodrigo Caetano, Conceptualization, Software, Formal analysis, Investigation, Methodology, Writing - original draft, Writing - review and editing; Yaroslav Ispolatov, Conceptualization, Data curation, Software, Formal analysis, Investigation, Visualization, Methodology, Writing - original draft, Writing - review and editing; Michael Doebeli, Conceptualization, Data curation, Formal analysis, Supervision, Funding acquisition, Investigation, Methodology, Writing - original draft, Project administration, Writing - review and editing

### Author ORCIDs
Rodrigo Caetano (ID) https://orcid.org/0000-0003-2837-113X
Yaroslav Ispolatov (ID) https://orcid.org/0000-0002-0201-3396
Michael Doebeli (ID) https://orcid.org/0000-0002-5975-5710

### Decision letter and Author response
Decision letter https://doi.org/10.7554/eLife.67764.sa1
Author response https://doi.org/10.7554/eLife.67764.sa2

## Additional files

### Supplementary files
• Transparent reporting form

### Data availability
All data generated or analysed during this study are obtained through the codes which have been deposited in https://github.com/jaros007/Codes_for_Evolution_of_diversity_in_metabolic_strategies (copy archived at https://archive.softwareheritage.org/swh:1:rev:d0a9ad7ca4459a1cc221b7bf1d1d311733400f0a).

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

## Appendix 1

### Ecological and evolutionary dynamics

We assume as in *Posfai et al., 2017* that metabolic reactions occur on a much faster time scale than cellular division, so that resource concentrations are always at their ecological equilibrium values $c_j^*$ determined as solutions of equations:

$$\frac{dc_j}{dt} = s_j - \left(\sum_\alpha n_\alpha(t)\alpha_j\right) r_j(c_j) - \mu_j c_j. \tag{A1}$$

with $dc_j/dt = 0$. (Note that these equilibrium resource concentrations are determined by the current populations sizes $n_\alpha(t)$.) In practice, a faster time scale can be achieved by multiplying the right-hand side of *Equation A1* by a large dimensionless constant. The cellular per capita birth rate $g_\alpha$ of species $\alpha$ is proportional to the amount of nutrient absorbed by each individual,

$$g_\alpha(c_1^*, \ldots c_p^*) = \sum_{j=1}^{p} \alpha_j r_j(c_j^*). \tag{A2}$$

The dynamics of the population size $n_\alpha$ then becomes

$$\frac{dn_\alpha}{dt} = \left(g_\alpha(c_1^*, \ldots c_p^*) - \delta\right) n_\alpha. \tag{A3}$$

To derive the evolutionary dynamics for the allocation strategies, that is, for the trait $\alpha$, we follow the adaptive dynamics approach, a powerful tool to study gradual evolutionary diversification due to frequency-dependent ecological interactions (*Metz et al., 1992*; *Dieckmann and Law, 1996*; *Doebeli, 2011*; *Dieckmann and Doebeli, 1999*; *Doebeli and Dieckmann, 2003*; *Geritz et al., 1997*; *Hui et al., 2018*). In particular, adaptive dynamics can generate the paradigmatic phenomenon of evolutionary branching (*Metz et al., 1992*; *Geritz et al., 1997*; *Doebeli and Dieckmann, 2000*; *Hui et al., 2018*; *Doebeli, 2011*; *Dieckmann and Doebeli, 1999*), during which a population that evolves in a continuous phenotype space first converges to a fitness minimum (evolutionary branching point) and then splits into two (or more) diverging phenotypic branches. We start with considering a monomorphic resident population at its ecological equilibrium $n_\alpha^*$, which is defined as the population size for which the equilibrium resource levels $c_j^*$ are such that $g_\alpha(c_1^*, \ldots, c_p^*) = \delta$ (note again that the $(c_1^*, \ldots, c_p^*)$ implicitly depend on $\alpha$). The invasion fitness of a rare mutant $\alpha'$ is then the per capita growth rate of the mutant $\alpha'$ at the resource levels defined by the resident:

$$f(\alpha, \alpha') = g_{\alpha'}(c_1^*, \ldots c_p^*) - \delta. \tag{A4}$$

To derive the adaptive dynamics, we consider the selection gradient $q(\alpha) = (q_1(\alpha), \ldots, q_p(\alpha))$, with components

$$q_i(\alpha) = \frac{\partial f(\alpha, \alpha')}{\partial \alpha_i'}\Big|_{\alpha'=\alpha}. \tag{A5}$$

$q(\alpha)$ defines a $p$-dimensional dynamical system in unrestricted $\alpha$-space,

$$\frac{d\alpha_i}{dt} = \sigma n_\alpha^* q_i(\alpha). \tag{A6}$$

The speed of evolution of $\alpha$ is proportional to the current ecological equilibrium population size $n_\alpha^*$ because the number of mutations occurring at any given point in time is proportional to $n_\alpha^*$. The parameter $\sigma$ describes both the per capita rate and effective size of mutations. Without loss of generality, we set $\sigma = 1$.

To take the enzymatic tradeoff into account, the unconstrained adaptive dynamics (*Equation A6*) needs to be restricted to the surface in $\alpha$-space that is defined by the tradeoff $\sum_{j=1}^{p} \alpha_j^\gamma = E$, where $E$ is a positive number (*Ito and Sasaki, 2016*). An illustrative example is the one in which the nutrients come from three different resources. The tradeoff $\alpha_1^\gamma + \alpha_2^\gamma + \alpha_3^\gamma = E$ defines a surface in $\alpha$-space

containing all strategies. The curvature of each surface is determined by $\gamma$. *Appendix 1—figure 1a* shows an example of the surface defined by the tradeoff for the case that $\gamma > 1$ while *Appendix 1—figure 1b and c* show the curvature for the case where $\gamma = 1$ and $\gamma < 1$, respectively. The blue star and the orange diamond illustrate possible position of the strategies in $\alpha$-space. The individuals with strategy indicated by the blue star uptake nutrients only from resource $s_1$, while the individuals with strategy indicated by the orange diamond uptake nutrients from all three resources.

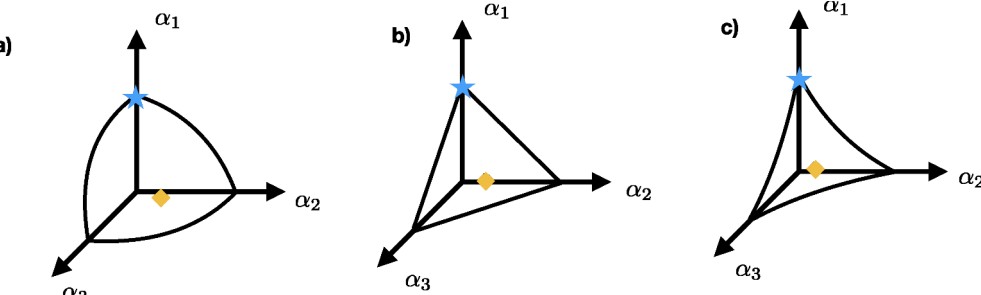

**Appendix 1—figure 1.** Three possible surfaces defined by a tradeoff: (**a**) shows the concave surface for the case $\gamma > 1$, while (**b**) and (**c**) show the surface for the cases $\gamma = 1$ and $\gamma < 1$, respectively. The blue star and the orange diamond represent possible strategies. Individuals with strategy represented by the blue star obtain their nutrients only from resource $s_1$ while the individuals that adopt strategy indicated by the orange diamond uptake nutrients from all three resources.

Equilibrium points of the adaptive dynamics, the so-called singular points, are resting points $\alpha^*$ of the resulting dynamical system in phenotype space. Given a singular point $\alpha^*$, two stability concepts are important. First, there is stability in the usual sense of converging to $\alpha^*$ from nearby initial conditions, which is measured by the Jacobian matrix of the functions defining the adaptive dynamics, evaluated at $\alpha^*$. Second, evolutionary stability is measured by the Hessian of the invasion fitness function $f(\alpha^*, \alpha')$ with respect to the mutant trait, evaluated at the singular point $\alpha^*$, and taken along the constraint surface (*Ito and Sasaki, 2016*). A negative definite Hessian (all eigenvalues negative) means that the singular point is a maximum of invasion fitness and no branching occurs. Alternatively, a singular point is called an evolutionary branching point if it is both convergent stable with regard to the Jacobian and evolutionarily unstable with regard to the Hessian. Thus, a singular point is a branching point if all eigenvalues of the Jacobian have negative real parts, and if the Hessian matrix is not negative definite.

## Singular points and their convergence and evolutionary stability

The case where the decay rates, $\mu_j$, are zero for any $j$ admits an analytical solution. We consider allocation strategies $\alpha = (\alpha_1, ..., \alpha_p)$ as in the main text, but here we assume a more general tradeoff function:

$$\sum_{j=1}^{p} b_j \alpha_j^{\gamma_j} = 1. \tag{A7}$$

It turns out to be convenient to reparametrize the strategy space as follows:

$$\beta_j \equiv b_j \alpha_j^{\gamma_j} \tag{A8}$$

for $j = 1, ..., p$. This simplifies the tradeoff expression to

$$\sum_{j=1}^{p} \beta_j = 1. \tag{A9}$$

Because the $\beta_j$ increase monotonically with $\alpha_j$, the adaptive dynamic properties in terms of convergence and evolutionary stability of singular points are the same for $\alpha = (\alpha_1, ..., \alpha_p)$ and $\beta =$

$(\beta_1, ..., \beta_p)$ phenotypes. However, the tradeoff in $\beta$, *Equation A9*, is linear, which simplifies the analysis.

In terms of $\beta$, the per capita rate of use of resource $j$ of an individual with phenotype $\beta$ is

$$\left(\frac{\beta_j}{b_j}\right)^{\frac{1}{\gamma_j}} r_j. \tag{A10}$$

We assume that nutrients are supplied to the system at a constant rate given by the vector $s = (s_1, s_2, \ldots, s_p)$, where $s_j$ is the supply rate of the $j$ th resource. We consider the low degradation rate regime, that is, $\mu_j \to 0$ for all $j$ in *Equation A1*. Setting the right-hand sides of *Equations A1, A3* equal to zero and taking into account that the sum in *Equation A1* consists of a single term, we obtain for the equilibrium density of a population monomorphic in $\beta$

$$n_\beta^* = \sum_{j=1}^{p} \frac{s_j}{\delta} \tag{A11}$$

The invasion fitness of a rare mutant with uptake strategy $\beta'$ in a resident $\beta$ at ecological equilibrium $n_\beta^*$ becomes

$$f(\beta, \beta') = \sum_{j=1}^{p} \left(\frac{\beta_j'}{\beta_j}\right)^{1/\gamma_j} \frac{s_j}{n_\beta^*} - \delta. \tag{A12}$$

To derive the adaptive dynamics of $\beta$, we calculate the selection gradient $q(\beta) = (q_1(\beta), ..., q_p(\beta))$ and project it onto the linear constraint space:

$$q_j(\beta) = \left.\frac{\partial f(\beta, \beta')}{\partial \beta_j'}\right|_{\beta' = \beta} = \frac{s_j}{\beta_j \gamma_j n_\beta^*} \tag{A13}$$

$$\frac{d\beta_j}{dt} = \sigma n_\beta^* \left(q_j(\beta) - \frac{1}{p}\sum_{k=1}^{p} q_k(\beta)\right) = \sigma n_\beta^* \left(\frac{s_j}{\beta_j \gamma_j n_\beta^*} - \frac{1}{p}\sum_{k=1}^{p} \frac{s_k}{\beta_k \gamma_k n_\beta^*}\right). \tag{A14}$$

Here, the term

$$\frac{1}{p}\sum_{k=1}^{p} q_k(\beta) = \frac{1}{p}\sum_{k=1}^{p} \frac{s_k}{\beta_k \gamma_k n_\beta^*} \tag{A15}$$

is the component of the selection gradient (*Equation A13*) that is orthogonal to the tradeoff hyperplane (note that $(1/\sqrt{p}, \ldots, 1/\sqrt{p})$ is a unit vector orthogonal to the tradeoff hyperplane).

If we set the mutational parameter $\sigma = 1$, the adaptive dynamics of $\beta$ becomes

$$\frac{d\beta_j}{dt} = \frac{s_j}{\beta_j \gamma_j} - \frac{1}{p}\sum_{k=1}^{p} \frac{s_k}{\beta_k \gamma_k}. \tag{A16}$$

Note that we only need $p - 1$ equations due to the (linear) tradeoff. It is easy to see that *Equation A16* has a unique fixed point, that is, there is a unique singular point for the adaptive dynamics given by

$$\beta_j^* = \frac{s_j/\gamma_j}{\sum_{k=1}^{p} s_k/\gamma_k} \tag{A17}$$

for $j = 1, ..., p$. In terms of the original trait $\alpha$, *Equation A17* is (*Equation 4*) in the main text.

To check for convergence stability of $\beta^* = (\beta_1^*, ..., \beta_p^*)$, we have to calculate the Jacobian matrix $J$ of the right-hand side of *Equation A16*, evaluated at the singular point $\beta^*$. It is easy to see that the $jk$ th element of $J$ is

$$J_{jk} = -\delta_{jk}\frac{s_j}{\gamma_j\beta_j^{*2}} + \frac{1}{p}\frac{s_k}{\gamma_k\beta_k^{*2}}. \tag{A18}$$

Thus, $J$ is of the form

$$J = J_d + A, \tag{A19}$$

where $J_d$ is a diagonal matrix with element $J_{jj} = -\frac{s_j}{\gamma_j\beta_j^{*2}}$ and $A$ is a matrix whose elements in the $k$ th column are all identical and equal to $\frac{1}{p}\frac{s_k}{\gamma_k\beta_k^{*2}}$. This implies that the matrix $A$ maps any vector in phenotype space to a vector that is orthogonal to the tradeoff hyperplane (i.e., to a multiple of the vector $(1,...,1)$). If $\Delta\beta = (\Delta\beta_1,...,\Delta\beta_p)$ is any vector of deviations from the singular point, it follows that the projection of $J\Delta\beta$ onto the tradeoff hyperplane is the same as the projection of the vector $J_d\Delta\beta$. Since all eigenvalues of $J_d$ are real and negative, it follows that the singular point $\beta^*$ is a local attractor, that is, convergent stable, regardless of the exponents $\gamma_j, j = 1,...,p$.

For evolutionary stability, we have to calculate the Hessian matrix $H$ of second derivatives of the invasion fitness function, *Equation A12* with respect to the mutant trait $\beta'$ and evaluated at the singular trait value $\beta^*$. The $jk$ th element of $H$ is

$$H_{jk} \equiv \frac{\partial^2 f}{\partial\beta_j'\partial\beta_k'}\bigg|_{\beta'=\beta=\beta^*} = \delta_{jk}\nu\frac{1-\gamma_j}{s_j}, \tag{A20}$$

where $\nu$ is a constant:

$$\nu = \delta\frac{\left(\sum_{l=1}^p s_l/\gamma_l\right)^2}{\sum_{m=1}^p s_m}. \tag{A21}$$

Thus, $H$ is diagonal (due to the transformation from $\alpha$ to $\beta$), and $H$ is negative definite, that is, all eigenvalues are negative, if and only if $\gamma_j > 1$ for all $j = 1,...,p$. Because the tradeoff hyperplane is linear in $\beta$, it follows that any index $j$ with $\gamma_j < 1$ provides a branching direction $\beta_j$, that is, a direction in phenotype space along which evolutionary diversification is possible. More precisely, any direction in $\beta$-space (other than orthogonal to the tradeoff surface) along which the unconstrained Hessian (*Equation A20*) has a minimum corresponds to a direction on the tradeoff surface along which diversification is possible.

The results presented in the main text now follow from the above by setting $\gamma_j = \gamma$ for $j = 1,...,p$. But the above analysis also suggests that with suitably chosen $\gamma_j$, it is possible to generate evolutionary branching in some directions, but not in others. This is illustrated in *Appendix 1—figure 2*. Our analysis and numerical procedure are applicable to an evolving system of populations with any number of resources. To facilitate visualization, in the following, we consider just three resources, so that because of the constraint, each population is characterized by two independent parameters $\alpha_i$. *Appendix 1—figure 2* illustrates diversification in the direction of $\alpha_3$ ($\gamma_3 < 1$) without diversification in the directions $\alpha_1$ and $\alpha_2$ ($\gamma_1, \gamma_2 > 1$).

*Appendix 1—figure 3* illustrates diversification in the directions $\alpha_1$ and $\alpha_2$ ($\gamma_1, \gamma_2 < 1$), with no diversification in $\alpha_3$ ($\gamma_3 > 1$).

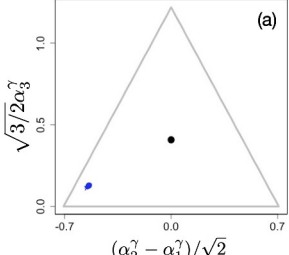 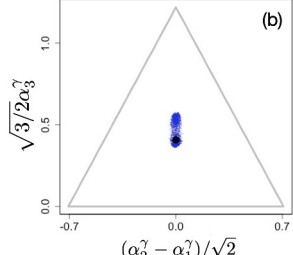 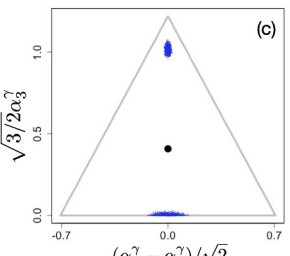

*Appendix 1—figure 2 continued on next page*

*Appendix 1—figure 2 continued*

**Appendix 1—figure 2.** Example of evolutionary dynamics for $\gamma_1 = \gamma_2 = 1.1$ and $\gamma_3 = 0.9$, showing convergence to the singular point and subsequent diversification only in the $\alpha_3$ direction. (Note that the dynamics are shown in the original $\alpha$-phenotype space.) The corresponding video can be found here, each frame in the video is separated by 2,000 time steps. Other parameter values were $s_j = 1$, $\mu_j K_j = 0.1$ for $j = 1, 2, 3$, and $\delta = 0.25$.

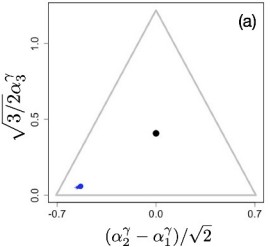 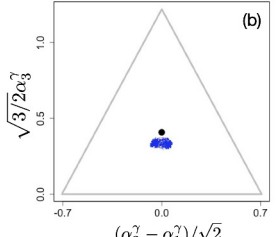 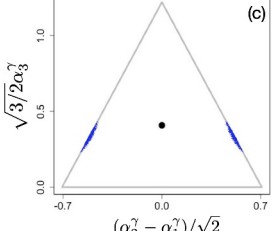

**Appendix 1—figure 3.** Example of evolutionary dynamics for $\gamma_1 = \gamma_2 = 0.9$ and $\gamma_3 = 1.2$, showing convergence to the singular point and subsequent diversification only in $\alpha_1 - \alpha_2$ directions. (Note that the dynamics are shown in the original $\alpha$-phenotype space.) The corresponding video can be found here, each frame in the video is separated by 2000 time steps. Other parameter values were $s_j = 1$, $\mu_j K_j = 0.1$ for $j = 1, 2, 3$, and $\delta = 0.25$.

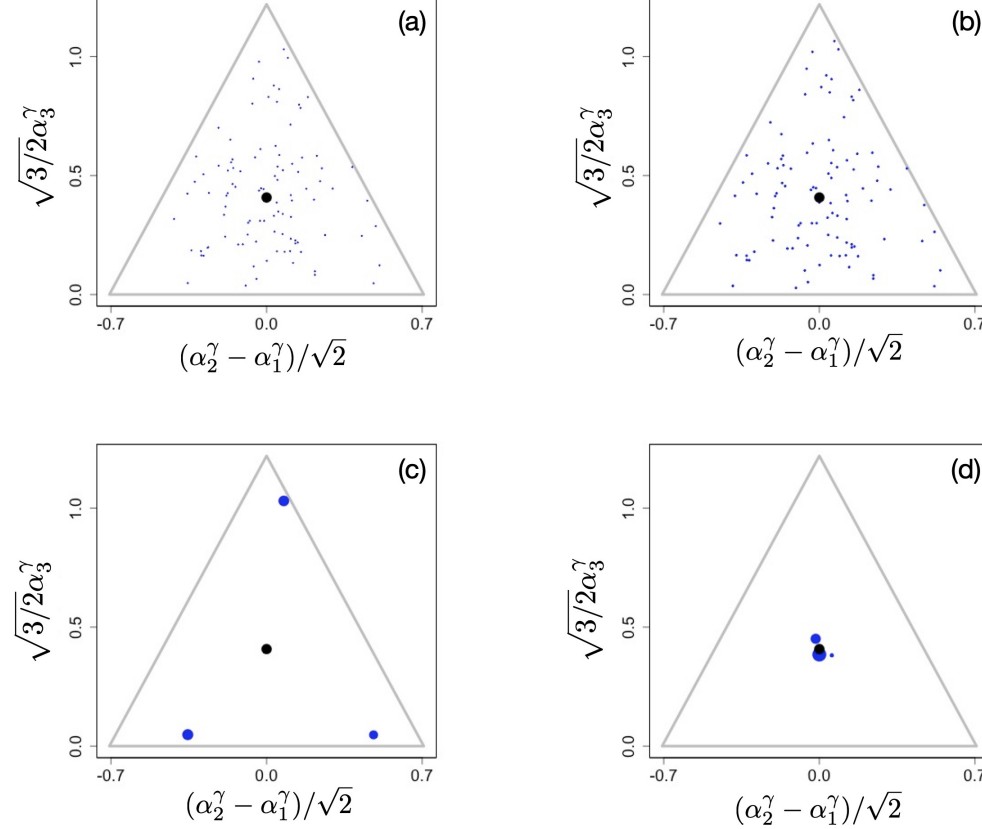

**Appendix 1—figure 4.** Initial population configuration of 100 randomly placed clusters in the phenotypic simplex (a), final configurations after 5,000,000 time units for $\gamma = 1$ (b), $\gamma = 0.9$ (c), and $\gamma = 1.1$ (d). Videos of the entire ecological processes can be found here, time interval between frames increased as a geometric progression, $t(i+1) = 1.05t(i)$. Other parameter values were $s_j = 1$, $\mu_j K_j = 0.1$ for $j = 1, 2, 3$, and $\delta = 0.25$.

## Numerical procedures

In the chemostat simulations, we numerically integrate the system of population dynamics *Equation A3* for $M$ populations using a simple Euler update ($M = 1$ at the beginning of the simulations). After each integration step, the populations that fall below a small 'extinction' threshold density (normally $n_{min} = 10^{-6}$) are removed from the system. The resource concentrations $c_i$ and uptake rates $r_i$ are considered relaxed to their steady states for a given set of populations $\{n_\alpha\}$,

$$r_i = \frac{(s_i + \phi_i + \mu_i K_i) - \sqrt{(s_i + \phi_i + \mu_i K_i)^2 - 4\phi_i s_i}}{2\phi_i}, \tag{A22}$$

where $\phi_i = \sum_\alpha n_\alpha \alpha_i$.

To simulate serial dilutions, we numerically integrate *Equations 6, 7* for $M$ populations and $p$ resources using also the Euler update ($M = 1$ at the beginning of the simulations). After each integration step, the populations that fall below a small 'extinction' threshold density (normally $n_{min} = 10^{-6}$) are removed from the system. Once the resources are depleted so that the condition (*Equation 8*) is satisfied, the populations of all existing species are rescaled according to *Equation 5* and the resource concentrations are reset to $c_j(0)$.

To mimic mutations in both simulation setups, periodically (typically once every $\Delta t_{mutation} = 1$ time units) a mutant is split from an ancestor, which is randomly chosen with probability proportional to

its total birth rate. The mutant's phenotype is randomly offset from the ancestral phenotype along the constraint surface. The offset distance is drawn from a uniform distribution in the interval $[-m, m]$. Unless otherwise noted, $m = 0.005$. The mutant population is set to be 10% of the ancestral one, and the ancestor population is reduced by 10%. In addition to mutations, periodically (typically once every $\Delta t_{merge} = 100\Delta t_{mutation}$ time units) populations that are within a distance $m$ of each other are merged (preserving their phenotypic center of mass) and their population sizes added. Periodic repetition of mutation and merging procedures preserves the phenotypic variance necessary for evolution while limiting computational complexity.

This produces clouds or clusters of $\alpha$-values in phenotype space (see figures), with each phenotype $\alpha$ representing a monomorphic population of individuals with that phenotype $\alpha$. Somewhat imprecisely, we refer to a distinct cluster of phenotypes as a species. The clusters move in phenotype space due to extinction and merging of phenotypes, and due to creation of new phenotypes by mutation. This movement represents evolution and occurs along the constraint surface. A diversification event occurs when a cluster corresponding to the diversifying species spontaneously splits into two or more clusters that diverge from each other and move apart.

## Maintenance of diversity in ecological time scales

Here, we briefly show how non-linear trade-offs affect diversity on ecological time scale. To do this, we initiate the simulations with a set of for example randomly chosen phenotypes throughout phenotype space and then run the systems with the mutational process turned off. In *Appendix 1—figure 4a*, we show the initial configuration used for three different scenarios with different exponents $\gamma$ of the tradeoff (here, we again assume that $\gamma_j = \gamma$ for all $j$). The functional form of the tradeoffs has a profound effect on the number of surviving species, with many species coexisting when $\gamma = 1$, as reported in *Posfai et al., 2017* (*Appendix 1—figure 4b*), but with typically only $p$ species surviving when $\gamma < 1$ (*Appendix 1—figure 4c*) and only very few species surviving in the close vicinity of the singular point when $\gamma > 1$ (*Appendix 1—figure 4d*).

Videos of ecological simulation of serial dilution scenario can be found here.

## Non-linear metabolic rates

Consider the consumption or transformation of a resource substance $c$ into a downstream metabolic product $m$ using a specific enzyme $\alpha$:

$$c + \alpha \underset{k_{-1}}{\overset{k_1}{\rightleftharpoons}} c\alpha \overset{k_2}{\longrightarrow} m + \alpha. \tag{A23}$$

The assumption that the concentration of the complex $c\alpha$ instantaneously relaxes to its steady state defined by the current concentrations of $c$ and $\alpha$ constitutes the Michaelis-Menten approximation,

$$\frac{\partial [c\alpha]}{\partial t} = 0. \tag{A24}$$

Assuming mass action kinetics and denoting by $c$ and $\alpha$ the total (bound in the complex plus free) concentrations of the resource and enzyme, and dropping the traditional symbols for concentrations, one gets a quadratic equation for the concentration $\psi$ of the $c\alpha$ complex:

$$k_1(c - \psi)(\alpha - \psi) = (k_2 + k_{-1})\psi, \tag{A25}$$

with the solution

$$\psi = \frac{\alpha + c + \chi - \sqrt{(\alpha + c + \chi)^2 - 4\alpha c}}{2}. \tag{A26}$$

Here, $\chi \equiv (k_{-1} + k_2)/k_1$ is the dissociation constant for the complex. The more common form of the Michaelis-Menten approximation,

$$\psi \approx \frac{\alpha c}{c + \chi},$$ (A27)

is obtained in the limit when the substrate concentration is much larger than that of the enzyme. (We note that this form is linear in $\alpha$ and is identical to the product of a Monod function and the corresponding $\alpha$ used in *Equation 3*). Since

$$\frac{\partial^2 \psi}{\partial \alpha^2} = -\frac{2c\chi}{\left[(\alpha - c + \chi)^2 + 4c\chi\right]^{3/2}} < 0 \text{ for any } \alpha > 0,$$ (A28)

the resource uptake and the growth rates are always sub-linear in the concentration of the enzyme $\alpha$.

