## [Decision Letter]

**Acceptance summary:**

This manuscript generalizes a recently-published framework by the Wingreen group which considers adaptation/evolutionary dynamics in a consumer resource model when a tradeoff on metabolic rates is imposed. The original model involved a linear tradeoff: e.g. the faster a microbe can eat nutrient 1, the slower it can eat nutrient 2, so that a sum-rule is imposed on the maximal rates. This paper considers the mathematics of the effect of nonlinear tradeoffs, in the form of a sum of rates raised to a power γ, with γ not just equaling 1 (γ=1 represents the regime of linear tradeoff). The authors find that, keeping all other model features the same, diversity is lost when γ is not 1. That is, nonlinear tradeoff destroys diversity.

**Decision letter after peer review:**

[Editors’ note: the authors submitted for reconsideration following the decision after peer review. What follows is the decision letter after the first round of review.]

Thank you for submitting your work entitled "Evolution of diversity in metabolic strategies" for consideration by *eLife*. Your article has been reviewed by 3 peer reviewers, including Wenying Shou as the Reviewing Editor and Reviewer #1, and the evaluation has been overseen by a Senior Editor .

We are sorry to say that, after consultation with the reviewers, we have decided that your work will not be considered further for publication by *eLife*.

Although one reviewer was positive, the overwhelming sentiment was that the work does not sufficiently advance the field.*Reviewer #1:*

Previous theoretical work argued that among species that compete for resources, physiological tradeoff (e.g. consuming more of food 1 leads to consuming less of food 2) can give rise to species diversity that greatly exceeds the number of resources, even in a well-mixed environment not conducive for species diversity. If previous work were to be general, it would be exciting because this offers a clue to the puzzle scientists have been trying to solve for a long time: what supports high species diversity? Caetano et al show that the finding from previous work only holds under a very restrictive condition (tradeoff function being linear). When that condition is violated (which can frequently occur in biology), we end up with either a single generalist species, or specialists each specializing on a single resource. Thus, in general, the total number of species cannot be larger than the total number of limiting resources in a well-mixed environment, as posited by the competitive exclusion principle. In short, we are back at where we were.

The authors started by introducing the adaptive dynamics framework which has been used to study evolutionary diversification due to frequency-dependent selection. The introduction is not adequate in giving nonspecialists an intuitive feeling about how evolutionary branching point from fitness minimum works. Of course, making theoretical conclusions accessible is not trivial, and may not be achievable. However, authors can try harder by including supplementary figures.

The authors then introduced work from the Wingreen group: R number of resources in a well-mixed environment, and each consumer species has an uptake strategy for each of the R resources. The total uptake for each species is fixed, creating a trade-off: more uptake of one resource reduces the uptake of another. Previous work showed that when the tradeoff function is linear (e.g. the sum of [uptake of each resource)^γ^] = a constant where the exponent γ=1, then many species with different uptake strategies can coexist.

The authors showed that the conclusion from previous work is rather restricted in its scope: large diversity only exists when the exponent γ in the tradeoff function is 1 (i.e. linear). When that exponent is greater than 1, no diversification occurs (saving for mutation-selection balance) and all individuals are phenotypically similar generalists. When the exponent is less than 1 (i.e. concave), the initial convergence to an unstable steady state later evolutionarily diverges into specialists, each specializing on a resource.

Authors also tested their conclusions within the ecological framework of the Wingreen group, and in different scenarios (chemostat versus serial dilutions), and reached the same conclusions.

The paper is relatively easy to read (in the realm of theoretical papers). This work reminds me of the work that has been done in the field of the evolution of cooperation. For example, public goods games often assume that the effect of the public good is a linear function of the number of contributions, an assumption that is often violated in biology. Depending on whether this function is linear or nonlinear, one can get very different outcomes in cooperator/non-cooperator coexistence (e.g. Archetti and Scheuring, JTB 299:9-20; Damore and Gore, JTB, https://doi.org/10.1016/j.jtbi.2011.03.008). Authors may want to add a discussion on that.

I wonder whether this paper should be added as a Research Advance to the original paper from the Wingreen paper published in *eLife*.

*Reviewer #2:*

This paper deals with the diversification of metabolic strategies in an evolving population. The authors consider a consumer-resource model under different metabolic trade-offs (sublinear, linear, and superlinear). They show that the linear case is a marginal scenario that corresponds to high diversity as a consequence of neutral evolution. Both the sub- and superlinear cases lead to the coexistence of fewer species than resources, as expected by the competitive exclusion principle.

The manuscript is well written and easy to follow. The derivation using adaptive dynamics is interesting and the results are robust. I am mainly concerned by the premises of the work.

l 70 "This is a very interesting finding because it violates the competitive exclusion principle". This is not strictly correct (and I think this is a central point). To violate the competitive exclusion principle, more species than resources should *stably* coexist. The stability requirement is essential. Otherwise, the principle can be easily falsified by a neutral model: in presence of even 1 single resource, an arbitrary number of ecologically equivalent species coexist (neutrally). Neutral coexistence is, as well known, structurally unstable: arbitrary small differences (which break the ecological equivalence) drive many species to extinction (restoring the bound on diversity given by the number of resources).

In the model by Posfai et al. (2017) coexistence is in fact only neutral. There is a manifold of fixed points and stability is marginal (several eigenvalues of the community matrix are equal to zero, e.g. see https://arxiv.org/pdf/2002.04358 ). The fixed point of their dynamics (abundance of different species) depends on the initial conditions. The high levels of diversity are, as a consequence, structurally unstable. This can be shown in multiple ways: introducing an (arbitrarily small) species variability in the trade-off (E depends on species identity), introducing variability in the dilution rate d (appearing in eq 3), or, as done in the paper, by altering the functional form of the trade-off.

This is a central point. It explains why many species are observed in the model by Posfai et al. And it explains why the result is extremely (infinitely) sensitive to the parameterization. These ecological considerations are mirrored in the fact that for gamma = 1 the evolutionary dynamics are neutral.

My main concern about this work is whether it has a sufficient degree of novelty and interest. As mentioned in the public review, the results are robust. But, after a close analysis of the model by Posfai et al., they are not unexpected. The manuscript, as it stands, mostly demonstrates the weaknesses of the paper by Posfai et al.: fixed points with more species than resources exist, but they are only marginally stable (https://arxiv.org/pdf/2002.04358 ). Structural instability is a direct consequence of this fact. Non-linear tradeoffs are just one (among many) ways to show that the results are infinitely sensitive to the parameter choices.

*Reviewer #3:*

There has been considerable recent interest in understanding the high degree of diversity observed in microbial communities. From a theoretical perspective, this has led to a resurgence of interest in resource-competition models. Several recent papers have studied the effects of trade-offs on total enzyme budgets within these models. An interesting observation is that with exact trade-offs, communities can self-organize to a state with an arbitrarily large number of species coexisting. One assumption of these models is that the total "cost" of enzymes is a linear function. The current work relaxes this assumption, and shows that this state of arbitrarily high coexistence relies on the linearity of the cost function.

Strengths: This study is rigorous, clearly presented, and the conclusions are mathematically sound. The authors analyze both chemostat and serial dilution systems.

Weaknesses: The results are qualitatively as expected from previous studies of the role of inexact trade-offs, and are more limited. The nonlinear trade-offs explored here are essentially equivalent to the unequal enzyme budgets explored in prior work. Indeed, these nonlinearities can be directly mapped to unequal budgets: for example, a nonlinearity that favors expression of a single enzyme is directly equivalent to a larger enzyme budget for species that produce only a single enzyme. Previous studies showed that unequal enzyme budgets lead to a loss of diversity, as is found in this work. Moreover, these prior studies found that even if trade-offs are not exact, the slow loss of diversity due to inexact trade-offs can be offset by invasion of new strategies and can therefore still lead to a large number of coexisting species.

The likely impact of this work on the field is modest, given that those who are already experts in the field will recognize that nonlinear trade-offs are equivalent to unequal enzyme budgets. Moreover, the current study does not actually provide any specific support for nonlinear trade-offs other than a few remarks in the Introduction.

The work would be of greater general interest if the biological evidence for nonlinearities in enzyme costs were carefully examined; mechanistic insights on how enzyme budget nonlinearities may arise in nature would be of significant utility to the field. However, this would require a substantial additional undertaking and would shift the focus of the work from the specific implications of nonlinearities in resource-competition models. An alternative would be to publish the current study in a more specialized journal, with a more theoretical focus.

It would also be helpful to provide a quantitative assessment of the sensitivity of diversity to the degree of nonlinearity. It is clear that any nonlinearity (or inexactness of trade-offs) leads to loss of diversity at long times. However, a small rate of invasion by new strategies can still lead to a diverse stationary state of the population. Given a certain degree of nonlinearity, how much invasion is required to maintain diversity? The adaptive dynamics calculations performed by the authors do not address this point because new strategies are only introduced if these are more fit than the residents. The question of diversity requires introducing invaders that may be slightly less fit, but still manage to survive due to demographic noise.

[Editors’ note: further revisions were suggested prior to acceptance, as described below.]

Thank you for submitting your article "Evolution of diversity in metabolic strategies" for consideration by *eLife*. Your article has been reviewed by 3 peer reviewers, including Wenying Shou as the Reviewing Editor and Reviewer #1, and the evaluation has been overseen by Aleksandra Walczak as the Senior Editor.

Essential Revisions:

Please consult the Reviewers' comments and address these in your revision.

*Reviewer #1:*

I remain supportive, especially if authors can discuss empirical measurements of biological tradeoffs and whether in the natural environment the linearity assumption might break down.

*Reviewer #3:*

The authors have made several cosmetic changes which have improved the clarity of their manuscript. However, the revised version does not substantially address my main concerns. The real question is whether the current manuscript makes a substantial contribution to the topic of microbial diversity. The focus of the paper is a critique of a model of resource competition with trade-offs. It is certainly legitimate to be critical of existing models. However, I believe the readers of *eLife* already appreciate the adage "all models are wrong, but some are useful". The authors have focused their attention on the first part of the adage, arguing that because growth functions will not be exactly linear the model is "wrong". But it's not news that the model is "wrong" (see above), the question is whether the model might still be a useful starting point for understanding diversity? What seems to me to be missing in the discussion, both in the original studies of Ref. 3 and 5 and the current manuscript, is quantification of how "wrong" the initial model is, and whether this undermines its utility. This is why I suggested that the authors carefully examine the "biological evidence for nonlinearities in enzyme costs". Their revised manuscript adds some sentences on this point, but in a non-quantitative way: the authors continue to make the mathematical point that the model is "wrong", but have not taken up the challenge of addressing whether it is or is not "useful". Yes, it is mathematically correct as the authors state that bimolecular reactions are strictly nonlinear in the reactants. But for typical enzyme concentrations in the μM range and typical metabolites in the 0.1-1 millimolar range, these nonlinearities are in the 0.1-1% range. From a biological perspective, a linear function might therefore still be a useful starting point. I've also read the references the authors cite on other sources of nonlinearity – they are equally non-quantitative. For example, the review by Marianayagam et al. states (without citations) "In its simplest form, oligomerization functions as a general mechanism for sensing protein concentration. An increase in protein concentration above the oligomerization threshold can be the stimulus for enzyme activation; similarly, enzyme deactivation will apply when cellular levels of the enzyme fall." I absolutely agree that for enzymes that need to oligomerize to function, this implies a mathematically nonlinear processing rate as a function of enzyme concentration. However, again for enzyme levels in the μM range and oligomerization dissociation constants in the commonly observed 1-10 picomolar range, the nonlinearities are again ~0.1-1%. Despite the revisions on this and other points in the reviews, in the end I am left still wondering whether the original model is "useful". My conclusion is that the current manuscript will be primarily of interest to researchers whose focus is on the mathematics of resource-competition models, and would therefore be appropriate for a more mathematically focused journal.

*Reviewer #4:*

This is a well-reasoned and well-presented paper, which considers adaptation/evolutionary dynamics in a consumer resource model when a tradeoff on metabolic rates is imposed. This finding will be of interest to the ecology community, and is well presented. Moreover, these consumer-resource models have relevance beyond the microbial ecology scenarios outlined here with applications in biology and beyond. Though the mathematics of their specific suggested model is well-analyzed, it stands to reason that when considering realistic systems, other timescales can be involved. As a result, the interplay between those timescales and other model parameters could quite possibly lead again to stable coexistence, as in the case of gamma=1.

The main caveat of this paper is that one does not need to resort to nonlinear tradeoffs to destroy diversity in the original Wingreen models, since simply giving different organisms different enzyme budgets would suffice - a plausible enough scenario. In the case of unequal enzyme budgets, by adding other (plausible) diversity stabilizing mechanisms it is possible to retain the coexistence state. Similarly, for values gamma sufficiently close to 1, I would expect that other realistic effects could again lead to a state resembling gamma=1. I would have liked to see this paper attempt to understand how diversity is maintained away from gamma=1, rather than just show us a way to break it.

The reality of the field (if I may) is that none of us have the answer to the so-called Paradox of the Plankton. There are several competing theories out there, and each of them has its merits and imperfections. Simply outlining an expected and obvious lack of generalizability of a particular aspect of an otherwise solid and informative theory is less constructive than this paper could be. Especially considering it is well known that consumer-resource models (e.g. Wingreen) support coexistence only on equal enzyme budgets unless other diversity stabilizing mechanisms are included.

What I think that the authors should focus on, is that neither the Wingreen papers nor recent papers from other groups claim that their particular framework captures all features of diversity maintenance. Instead, we are all trying, as a community, to piece together a coherent theory, stumbling along as we inevitably do. For example, the Wingreen papers consider violation of the linear constraint (i.e. unequal enzyme budgets) in several stochastic and deterministic cases, outlining their relevance. There are several timescales at play here and one can think how their interplay with particular model parameters influences the resulting steady state. This is likely the case with regards to this manuscript. Already, if one considers small deviations from γ = 1, one introduces a new timescale, to compete with other model timescales. Moreover, the "mutation" dynamics (Eq. 14) introduce one or more timescales, which are conveniently swept under the rug with a "without loss of generality" (l463), a statement without actual backing as far as I understood. Furthermore, the authors do not sufficiently acknowledge that their suggested "mutation" dynamics are a very specific simplification of true dynamics. Let us agree that true dynamics again include many timescales and confounding factors and that therefore this manuscript is no less fine-tuned than the theory it challenges. What the authors essentially do is, impose specific dynamics and then demonstrate that these specific dynamics fail to maintain diversity at . It seems likely that their "mutation" dynamics can be plausibly modified so that diversity is regained. After all, natural ecosystems do involve resource competition and the plankton are diverse.

Reading the other referee reports, it appears that some referees do not consider this paper sufficiently surprising to be published in *eLife*. However, one referee suggested that it be published as a Research Advance, to the original Wingreen paper published in *eLife*. I think that if the authors sufficiently improve this manuscript, the Research Advance track might make sense. Specifically, what I suggest, is for the authors to re-adjust their focus. Their results are well argued but simply show a known weakness in the theory – a weakness that can overcome. Why not argue them in a way that seeks to expand the field? The authors raise an interesting question, what needs to be added to the plain-vanilla version of consumer-resource models so that diversity is regained despite (slightly) nonlinear tradeoffs and a specific form of "mutation" dynamics? More elaborate model variants already incorporate other diversity stabilizing mechanisms which might well maintain diversity at non-unity values of γ, e.g. a recent preprint by Huang group in Stanford https://www.biorxiv.org/content/10.1101/2021.05.13.444061v1. Should the authors answer this question, and salvage diversity from nonlinear tradeoffs, I think this manuscript will be much improved. Exploring an example whereby diversity is re-instated in the consumer-resource framework (despite nonlinear tradeoffs) would demonstrate how in fact, suitably adjusted, consumer-resource models can be used to capture such competing ecological forces. I believe that by steering their manuscript to open new avenues of thought rather than closing avenues of thought, it would promote future inquiries in the field, hopefully ultimately leading to a deeper understanding. With this suitable addition and adjustment of the manuscript, I hope that the Editor and other referees would then agree that it would pass the threshold of contribution to be included as a Research Advance.

2. If indeed this paper is to be considered to be published in *eLife* as a Research Advance following the Wingreen *eLife* paper, it is a good idea that the authors change their notation to match precisely that parent paper's notations. I understand this is an unusual request but I do believe that it would serve future readers best – to smoothly carry over notations from one paper to its immediate follow-up in the same journal. As it is, the notation differences between the two papers are small and this is not a big request. Future readers will thank you.

---

## [Author Response]

[Editors’ note: The authors appealed the original decision. What follows is the authors’ response to the first round of review.]

Reviewer #1:The authors started by introducing the adaptive dynamics framework which has been used to study evolutionary diversification due to frequency-dependent selection. The introduction is not adequate in giving nonspecialists an intuitive feeling about how evolutionary branching point from fitness minimum works. Of course, making theoretical conclusions accessible is not trivial, and may not be achievable. However, authors can try harder by including supplementary figures.

We re-arranged the text, moving the general description of the adaptive dynamics to

the Appendix and focusing more on the predictions derived from both the numerical simulation of the complete model and the adaptive dynamics treatment of its approximate version. We also tried to better explain the convergence and evolutionary stability of the fixed point. This made the corresponding part of the Appendix a fairly complete verbal description of the standard adaptive dynamics procedure, illustrated by the equations pertinent to our model. However, given that the adaptive dynamics framework is almost 30 years old, and that many of the original and review papers on adaptive dynamics are already cited in the manuscript (there are thousands of publications based on adaptive dynamics), we stayed short of retelling a more detailed description of the general adaptive dynamics procedure.

The authors then introduced work from the Wingreen group: R number of resources in a well-mixed environment, and each consumer species has an uptake strategy for each of the R resources. The total uptake for each species is fixed, creating a trade-off: more uptake of one resource reduces the uptake of another. Previous work showed that when the tradeoff function is linear (e.g. the sum of [uptake of each resource)^γ^] = a constant where the exponent γ=1, then many species with different uptake strategies can coexist.The authors showed that the conclusion from previous work is rather restricted in its scope: large diversity only exists when the exponent γ in the tradeoff function is 1 (i.e. linear). When that exponent is greater than 1, no diversification occurs (saving for mutation-selection balance) and all individuals are phenotypically similar generalists. When the exponent is less than 1 (i.e. concave), the initial convergence to an unstable steady state later evolutionarily diverges into specialists, each specializing on a resource.Authors also tested their conclusions within the ecological framework of the Wingreen group, and in different scenarios (chemostat versus serial dilutions), and reached the same conclusions.The paper is relatively easy to read (in the realm of theoretical papers). This work reminds me of the work that has been done in the field of the evolution of cooperation. For example, public goods games often assume that the effect of the public good is a linear function of the number of contributions, an assumption that is often violated in biology. Depending on whether this function is linear or nonlinear, one can get very different outcomes in cooperator/non-cooperator coexistence (e.g. Archetti and Scheuring, JTB 299:9-20; Damore and Gore, JTB, https://doi.org/10.1016/j.jtbi.2011.03.008). Authors may want to add a discussion on that.

We thank the Reviewer for bringing up the connection between these two fields. Not

only the curvature of the benefit and cost functions play essential roles in the evolution of cooperation, the biochemical justifications of non-linearity in these functions are equally applicable to the case of non-linear uptake rates and trade-offs used in our work. We added the suggested references and a short explanation to the Discussion.

I wonder whether this paper should be added as a Research Advance to the original paper from the Wingreen paper published in eLife.

We agree that this would be the best place to publish our work.

Reviewer #2:My main concern about this work is whether it has a sufficient degree of novelty and interest. As mentioned in the public review, the results are robust. But, after a close analysis of the model by Posfai et al., they are not unexpected. The manuscript, as it stands, mostly demonstrates the weaknesses of the paper by Posfai et al.: fixed points with more species than resources exist, but they are only marginally stable(https://arxiv.org/pdf/2002.04358). Structural instability is a direct consequence of this fact. Non-linear tradeoffs are just one (among many) ways to show that the results are infinitely sensitive to the parameter choices.

As we stated at the beginning of this letter, we did our best to find any publication that

considers the models suggested in3,5, reports their structural instability, and presents the evolutionary and ecological results in the two generic scenarios of concave and convex trade-offs. We have not found any, and therefore we believe that such a publication is in order. We agree that there could be many other mechanisms that break the neutral evolutionary scenario and reduce the number of coexisting species to the competitive exclusion limit. Our goal in this work is to consider the interesting metabolic models developed in (3, 5) preserving their mechanistic definitions, yet supplying them with more realistic functional forms of trade-offs and nutrient uptake rates.

Reviewer #3:The work would be of greater general interest if the biological evidence for nonlinearities in enzyme costs were carefully examined; mechanistic insights on how enzyme budget nonlinearities may arise in nature would be of significant utility to the field. However, this would require a substantial additional undertaking and would shift the focus of the work from the specific implications of nonlinearities in resource-competition models. An alternative would be to publish the current study in a more specialized journal, with a more theoretical focus.

In short, the fundamental reason for non-linearities in trade-offs (enzyme costs) and uptake rates, which we showed to be interchangeable, is the non-linear dependence of chemical kinetics on the concentrations of reactants. A more detailed explanation is presented at the beginning of this letter. We added paragraphs to the Introduction and Discussion that present and discuss the evidence for non-linearity.

It would also be helpful to provide a quantitative assessment of the sensitivity of diversity to the degree of nonlinearity. It is clear that any nonlinearity (or inexactness of trade-offs) leads to loss of diversity at long times. However, a small rate of invasion by new strategies can still lead to a diverse stationary state of the population. Given a certain degree of nonlinearity, how much invasion is required to maintain diversity? The adaptive dynamics calculations performed by the authors do not address this point because new strategies are only introduced if these are more fit than the residents. The question of diversity requires introducing invaders that may be slightly less fit, but still manage to survive due to demographic noise.β−δβ

To give credit to adaptive dynamics, it does provide some estimates to how advantageous or disadvantageous ecological conditions are for an invader or a mutant. A selection gradient (e.g. (21-22)) is an indicator of the steepness of the fitness landscape, i.e. how quickly growth rate advantages or disadvantages change with the phenotypic separation from the resident population. Likewise, the Hessian of the invasion fitness (28) serves as an estimate of the second-order effects of phenotypic separation on the relative fitness. For example, the presence of 1− ƴ factor in (28) indicates that the strength of deviation from neutral selection increases in the leading order linearly with the deviation of the trade-off exponent ƴ from one.

Some information about the fate of an invader can be obtained from the videos corresponding to Figure 3 in the Appendix. Those simulations are initiated with a dense species packing and the majority of species quickly go extinct. To show that the effect is pronounced even for slight deviation from non-linearity, we chose the values for the trade-off exponent close to one, ƴ = 0.9 and ƴ = 1.1.

Studying the extinction rates of invaders under the conditions of the models (3, 5) could be an interesting separate project. In the present manuscript we prefer to focus on the steady state of evolving diversity.

References

[1] J. A. Borghans, R. J. De Boer, and L. A. Segel. Extending the quasi-steady state approximation by changing variables. *Bulletin of mathematical biology*, 58(1):43–63, 1996.

[2] A. Ciliberto, F. Capuani, and J. J. Tyson. Modeling networks of coupled enzymatic

reactions using the total quasi-steady state approximation. *PLoSComput Biol*, 3(3):e45,

2007.

[3] A. Erez, J. G. Lopez, B. G. Weiner, Y. Meir, and N. S. Wingreen. Nutrient levels and

trade-offs control diversity in a serial dilution ecosystem. *eLife*, 9:e57790, sep 2020.

ISSN 2050-084X.

[4] C. M. Hill, R. D.Waightm, andW. G. Bardsley. Does any enzyme follow the michaelismenten equation? *Molecular and cellular biochemistry*, 15(3):173–178, 1977.

[5] A. Posfai, T. Taillefumier, and N. S. Wingreen. Metabolic trade-offs promote diversity

in a model ecosystem. *Phys. Rev. Lett.*, 118:028103, Jan 2017.

[Editors’ note: what follows is the authors’ response to the second round of review.]

Essential Revisions:Please consult the Reviewers' comments and address these in your revision.Reviewer #1:I remain supportive, especially if authors can discuss empirical measurements of biological tradeoffs and whether in the natural environment the linearity assumption might break down.

Following the recommendation of Reviewer 1, in the new version of the manuscript we expanded the Introduction, Results and Discussion sections (see text in blue in the accompanying document). Assessing the empirical evidence for non-linearity in trade-offs and uptake rates, we focused on similarity in concentrations of metabolic enzymes and substrates, and on measured reaction elasticities and flux control coefficients as the most direct quantitative indicators of the “degree” of non-linearity. We also emphasized that our work leads to empirically testable predictions, at least in principle: microbial ecosystems with trade-off-structures that are close to linear should tend to be more diverse than those with highly non-linear trade-off structures (see new paragraph in the Results section). Finally, we propose possible ways to find a solution to the paradox of plankton based on non-stationary endogenous population dynamics (see new paragraph in the Discussion section).

Reviewer #3:The authors have made several cosmetic changes which have improved the clarity of their manuscript. However, the revised version does not substantially address my main concerns. The real question is whether the current manuscript makes a substantial contribution to the topic of microbial diversity. The focus of the paper is a critique of a model of resource competition with trade-offs. It is certainly legitimate to be critical of existing models. However, I believe the readers of eLife already appreciate the adage "all models are wrong, but some are useful". The authors have focused their attention on the first part of the adage, arguing that because growth functions will not be exactly linear the model is "wrong". But it's not news that the model is "wrong" (see above), the question is whether the model might still be a useful starting point for understanding diversity? What seems to me to be missing in the discussion, both in the original studies of Ref. 3 and 5 and the current manuscript, is quantification of how "wrong" the initial model is, and whether this undermines its utility. This is why I suggested that the authors carefully examine the "biological evidence for nonlinearities in enzyme costs". Their revised manuscript adds some sentences on this point, but in a non-quantitative way: the authors continue to make the mathematical point that the model is "wrong", but have not taken up the challenge of addressing whether it is or is not "useful". Yes, it is mathematically correct as the authors state that bimolecular reactions are strictly nonlinear in the reactants. But for typical enzyme concentrations in the μM range and typical metabolites in the 0.1-1 millimolar range, these nonlinearities are in the 0.1-1% range. From a biological perspective, a linear function might therefore still be a useful starting point. I've also read the references the authors cite on other sources of nonlinearity – they are equally non-quantitative. For example, the review by Marianayagam et al. states (without citations) "In its simplest form, oligomerization functions as a general mechanism for sensing protein concentration. An increase in protein concentration above the oligomerization threshold can be the stimulus for enzyme activation; similarly, enzyme deactivation will apply when cellular levels of the enzyme fall." I absolutely agree that for enzymes that need to oligomerize to function, this implies a mathematically nonlinear processing rate as a function of enzyme concentration. However, again for enzyme levels in the μM range and oligomerization dissociation constants in the commonly observed 1-10 picomolar range, the nonlinearities are again ~0.1-1%. Despite the revisions on this and other points in the reviews, in the end I am left still wondering whether the original model is "useful". My conclusion is that the current manuscript will be primarily of interest to researchers whose focus is on the mathematics of resource-competition models, and would therefore be appropriate for a more mathematically focused journal.

We also appreciate George Box’s ”All models are wrong, but some are useful” quote and frequently use it in teaching to motivate students to aspire to design useful models.

However, there are different kinds of ”wrong” in modeling. There is Box’s wrong, which

applies to all models, as they necessarily always represent a simplification of reality. But

there is also a more technical, and arguably more important model failure, which occurs

when a model is structurally unstable, which essentially means that if a particular model parameter is changed by a very small amount, the main quantitative and qualitative behaviour of the model changes completely. This is the case for Wingreen’s model with regard to the model parameter, which was set to 1 in Wingreen’s paper. As we show in our contribution, the model outcomes completely change when ƴ≠ 1 (no matter how large or small the deviation from 1 is, see below!). Thus, Wingreen’s model is wrong in the sense that it is structurally unstable. To suggest that our model is equally ”wrong” misses the point of structural instability.

Regarding usefulness, we maintain that generally assuming linearity of tradeoff appears

to be wrong biologically. That is, we argue that Wingreen’s model is too restrictive to be useful. Assuming that George Box’s statement was primarily related to his main field, statistics, we further support this claim by statistical arguments, as follows.

While we agree that there are metabolites, particular in fast-growing bacteria, with concentrations well in the millimolar range, many common metabolite have concentrations in the range of tens and even single digits of micromoles. For example, a detailed study (2) of the canonical model microbe *E. coli* during exponential growth in glucose, glycerol, or acetate as the carbon source revealed that out of 103 metabolites, 35 have concentrations above 1mM, but the concentrations of 46 metabolites are in tens or single micromole digits, including 2 metabolites with concentrations below 1 μM (as summarized in Table 1 in that paper). The database BIONUMBERS7 estimates the typical metabolite concentration in an *E. coli* bacterium as 32 μM, and provides evidence for concentrations of important *E. coli* glycolysis enzymes in tens and even hundreds of μM. Hence, while it looks natural that concentrations of metabolic substrates are larger than those of the corresponding enzymes, the difference is commonly few- to ten-fold rather than 2-3 orders of magnitude, as suggested by Reviewer 3.

A similar mischaracterization appears in the Reviewer’s estimate of the degree of nonlinearity caused by the disparity between the “oligomerization dissociation constants in the commonly observed 1-10 picomolar range” (”picomolar” is apparently rarely heard of, perhaps a typo in the Reviewer comments?), and concentrations of their monomer constituents. In reality, the difference between these quantities does not seem to be nearly as pronounced as the reviewer claims, as can e.g. be seen from the following summarizing quote in the review (1), page 5014: “The association between subunits can vary in strength and duration. Some proteins are found only, or primarily, in the oligomeric state. These proteins generally have dissociation constants in the nanomolar range. Others have a weak tendency to associate, with oligomerization dependent on environmental conditions, such as concentration, temperature, and pH. Such proteins often have higher Kd values in the μM or even millimolar range. Still other proteins oligomerize dynamically in response to a stimulus, such as a change in nucleotide binding, nucleotide hydrolysis or phosphorylation state. Such a change can have a dramatic effect on the affinity of the subunits for one another, often by orders of magnitude.”

Another review (15) discusses enzymes that are known to regulate their in vivo catalytic activity by dimeric and oligomeric association-dissociation, which rules out saturated binding between monomers and a corresponding linear dependence between the concentrations of monomers and complexes.

In addition to the comparison between enzyme and substrate concentrations and enzyme dissociation constants (tricky to measure in vivo even today, e.g.5), there is a more direct way to show omnipresence of non-linearity in metabolic reactions and fluxes. The Reviewer’s assessment that ”these nonlinearities are in the 0.1-1% range” probably refers to the definition of the degree of non-linearity of a function f(x) as a double-logarithmic derivative, d ln[(f(x))]/d ln(x). For a general power-law function, such as f(x) ≡ Cx that we used to define tradeoffs, the log-log derivative is equal to, the non-linearity parameter. For example, the log-log derivatives with respect to enzyme concentration of the tradeoff used in8 are equal to one. This derivative is equal to the local value of an exponent from an approximation (also local) of the function f(x) by a power law, and has long been used in metabolic analysis, being called an “elasticity coefficient” when f(x) is a particular reaction rate and x is the concentration of enzyme, or a flux control coefficient when f(x) is a flux through a metabolic pathway and x is again the concentration of enzyme. Given that we are more interested in the effect of concentration of enzymes on the growth rate, the latter definition seems more relevant.

As with metabolic control analysis and metabolic engineering, the mere existence of

these terms indicates that their values are often distinct from one. Indeed, such is the case in publications that present experimentally measured values of these coefficients, or quantities that are derived from those coefficients. For example, flux control coefficients of metabolic enzymes and plots of fluxes vs. enzyme concentrations shown in Figure 6 in (6), Table 6 in (4), Table 2 in (14), Figure 2 in (11), Figure 1 in (9), Table 5 in (10), Figure 2B in (13), Table 3 in (16) reveal that these coefficients rarely come close to 1, and that the dependencies of metabolic fluxes on enzyme concentrations are significantly nonlinear. (We found these and many other examples using Google Scholar to search for Flux Control Metabolic Coefficients Bacteria.) Indeed, back in the 60s Michael Savageau suggested to parametrize rates of complex enzymatic reactions as products of power-law functions of concentrations of enzymes and substrates12. This idea developed a substantial following, which once again indicates the necessity to account for non-linearity in kinetics of enzymatic pathways. Overall, the 0.1-1% range for nonlinearities seems doubtful.

In conclusion, it seems highly questionable to simply assume that all biologically relevant tradeoffs are close to linear. Rather, it seems very plausible that many of them are highly non-linear. To argue that readers of *eLife* would only need to know what happens in generic consumer-resource models when tradeoffs are linear therefore does not make sense to us. On the contrary, we think that readers of Wingreen’s papers, including their *eLife* paper, should be made aware of the structural instability of their results. We emphasize that because of the structural instability, tradeoffs would have to be exactly linear to arrive at Wingreen’s results, as any slight deviation from linearity destroys the neutrality necessary for excess diversity in Wingreen’s model, and hence results in completely different outcomes. This can be seen in (Author response image 1) which shows that even for ƴ = 0.99, i.e., for ƴ within 1% of the linear case, there is no excess diversity. The case for ƴ = 1.01 is shown in (Author response image 1) and the full evolutionary dynamics for this case can be found at https://figshare.com/s/f65ed0bf9b4305e9018f.

In the updated text, we substantially edited the Introduction and Discussion to point to

evidence in favour of the importance of non-linear kinetics and tradeoffs, and to reflect the above considerations. We also added a paragraph to the Results section discussing systems with different degrees of non-linearity and the expectations regarding the amount of diversity that can be derived based on the degree of non-linearity in tradeoff structures.

**Author response image 1. respfig1:** Snapshots illustrating the end of the evolutionary dynamics for (a) = 0. 99 and (b) = 1.01. The video of the entire evolutionary process can be found at https://figshare.com/s/f65ed0bf9b4305e9018f .

Reviewer #4:The reality of the field (if I may) is that none of us have the answer to the so-called Paradox of the Plankton. There are several competing theories out there, and each of them has its merits and imperfections. Simply outlining an expected and obvious lack of generalizability of a particular aspect of an otherwise solid and informative theory is less constructive than this paper could be. Especially considering it is well known that consumer-resource models (e.g. Wingreen) support coexistence only on equal enzyme budgets unless other diversity stabilizing mechanisms are included.

Again, this seems strange to us. It is exactly papers like ours that go some way to clarify the situation that a general model with non-linear tradeoff does not “support coexistence only on equal enzyme budgets ” (assuming that it refers to excessive diversity), but the Reviewer seems to think that all this is already well known. As we show, Wingreen’s models are structurally unstable, and therefore not ”solid and informative”, as the Reviewer claims. To us, it seems important that this is pointed out in the literature.

What I think that the authors should focus on, is that neither the Wingreen papers nor recent papers from other groups claim that their particular framework captures all features of diversity maintenance. Instead, we are all trying, as a community, to piece together a coherent theory, stumbling along as we inevitably do. For example, the Wingreen papers consider violation of the linear constraint (i.e. unequal enzyme budgets) in several stochastic and deterministic cases, outlining their relevance.

As far as we can tell, both papers coming from Wingreen’s group regarding this topic assume linear tradeoffs. We are unsure why the reviewer claims otherwise. We agree that we are all trying, and that all models and approaches have their own benefits and shortcomings. However, if a model has a particularly severe shortcoming, such that the

results simply don’t withstand scrutiny, this needs to be pointed out. In our opinion, this is the case for the structural instability due to assuming linear tradeoffs in Wingreen’s models.

There are several timescales at play here and one can think how their interplay with particular model parameters influences the resulting steady state. This is likely the case with regards to this manuscript. Already, if one considers small deviations from γ=1, one introduces a new timescale, to compete with other model timescales. Moreover, the "mutation" dynamics (Eq. 14) introduce one or more timescales, which are conveniently swept under the rug with a "without loss of generality" (l463), a statement without actual backing as far as I understood. Furthermore, the authors do not sufficiently acknowledge that their suggested "mutation" dynamics are a very specific simplification of true dynamics. Let us agree that true dynamics again include many timescales and confounding factors and that therefore this manuscript is no less fine-tuned than the theory it challenges. What the authors essentially do is, impose specific dynamics and then demonstrate that these specific dynamics fail to maintain diversity at . It seems likely that their "mutation" dynamics can be plausibly modified so that diversity is regained. After all, natural ecosystems do involve resource competition and the plankton are diverse.

Introducing ƴ≠1 does not introduce new timescales. It is true that our evolutionary framework does introduce a new (evolutionary) timescale that is dictated by mutations. However, this merely serves to generalize the model, and it does not affect the basic finding of the effect of non-linear tradeoffs on diversity. This can be seen by considering purely ecological dynamics, as happens in our models either at the evolutionary end state (when there is no evolutionary change anymore), or by simply setting mutations to 0, as we have done in the supplementary material. The resulting ecological dynamics show exactly the same effect of non-linear tradeoffs on diversity as the more general evolutionary model.

Thus, it is not the evolutionary time scale that causes these effects. As an aside: what we

meant by the phrase “Without loss of generality, we set σ = 1” quoted by the Reviewer, is that the constant σ, which essentially describes the rate and size of mutations, can be absorbed by defining a new time t’ = σt, which simply affects the frame rate in our videos, rather than what is in those frames.

Reading the other referee reports, it appears that some referees do not consider this paper sufficiently surprising to be published in eLife. However, one referee suggested that it be published as a Research Advance, to the original Wingreen paper published in eLife. I think that if the authors sufficiently improve this manuscript, the Research Advance track might make sense. Specifically, what I suggest, is for the authors to re-adjust their focus. Their results are well argued but simply show a known weakness in the theory – a weakness that can overcome. Why not argue them in a way that seeks to expand the field? The authors raise an interesting question, what needs to be added to the plain-vanilla version of consumer-resource models so that diversity is regained despite (slightly) nonlinear tradeoffs and a specific form of "mutation" dynamics? More elaborate model variants already incorporate other diversity stabilizing mechanisms which might well maintain diversity at non-unity values of γ, e.g. a recent preprint by Huang group in Stanford https://www.biorxiv.org/content/10.1101/2021.05.13.444061v1. Should the authors answer this question, and salvage diversity from nonlinear tradeoffs, I think this manuscript will be much improved. Exploring an example whereby diversity is re-instated in the consumer-resource framework (despite nonlinear tradeoffs) would demonstrate how in fact, suitably adjusted, consumer-resource models can be used to capture such competing ecological forces. I believe that by steering their manuscript to open new avenues of thought rather than closing avenues of thought, it would promote future inquiries in the field, hopefully ultimately leading to a deeper understanding. With this suitable addition and adjustment of the manuscript, I hope that the Editor and other referees would then agree that it would pass the threshold of contribution to be included as a Research Advance.

Once again, we are a bit perplexed by these comments. We tried to be succinct in our presentation of the fact that Wingreen’s model is structurally unstable, and that with

non-linear tradeoffs, no excess diversity can be maintained. Despite succinctness, this resulted in a full-fledged paper. Now the Reviewer wants us to write an additional paper about possible mechanisms that can lead to excess diversity with non-linear tradeoffs. This seems strange. Shouldn’t it be one paper at a time? Moreover, despite what the Reviewer may think, it is not clear at all at this point what possible mechanisms could indeed lead to excess diversity with non-linear tradeoffs. One possibility would be to consider non-equilibrium ecological dynamics. This has been done in the *eLife* paper from the Wingreen group to which our paper refers. In that paper, the authors considered seasonal ”batch culture” dynamics rather than equilibrium consumer-resource dynamics. However, they still always assumed linear tradeoffs! Note that this is a paper all of its own, and not merely an additional section added on to their original PRL paper…how could it be otherwise? We note that such batch culture dynamics do not lead to excess diversity with non-linear tradeoffs, as we have shown in our manuscript.

We are equally keen to find a solution to the Paradox of Plankton, and it is possible that

non-equilibrium ecological dynamics can allow for the maintenance of excess diversity, as we have recently shown using a different ecological model3. More precisely, we show that endogenous non-stationary “boom-bust” population dynamics can lead to a few-fold increase in diversity above the saturation limit expected with equilibrium population dynamics. This together with many experimental results reporting non-stationarity and apparent chaoticity of the population dynamics of actual plankton species makes us believe that the key to explain the astounding diversity of species is not the neutral evolutionary regime predicted in 8, but rather non-stationary population dynamics induced by competition and predation and perhaps external factors. We think that the neutral evolutionary state and linear tradeoff to 9 which an evolving system may accidentally “self-organize” is very fragile and at most temporal and cannot play a major role in establishing and maintaining (even less likely!) excess diversity.

However, it seems obvious that to extend investigations of non-equilibrium ecological

dynamics with non-linear tradeoffs to Wingreen’s consumer-resource model would need a new research project that would require its own space, and cannot be added to a paper about the structural instability of Wingreen’s model. The preprint cited by the Reviewer also proves the point: such investigations must be done in their own right, not tacked on to something else. Clearly, we must leave the exploration of excess diversity with non-linear tradeoffs to the future.

We have added a paragraph to the Discussion where we bring up the possibility of exploring the effect of non-linear tradeoffs in the presence of non-equilibrium ecological dynamics.

2. If indeed this paper is to be considered to be published in eLife as a Research Advance following the Wingreen eLife paper, it is a good idea that the authors change their notation to match precisely that parent paper's notations. I understand this is an unusual request but I do believe that it would serve future readers best – to smoothly carry over notations from one paper to its immediate followup in the same journal. As it is, the notation differences between the two papers are small and this is not a big request. Future readers will thank you.

There seem to be four discrepancies between our and Wingreen’s notation:

the number of resources p in 8 vs R (us), resource input rates in 8 vs S (us), death rate _ in8 vs d (us), and a dedicated summation index _ in sums over all species in 8, which is absent in our paper, because it is not needed. We have fixed the first three discrepancies, making our notation identical to that in 8. As for the summation index, we think that our notation is more succinct, less cumbersome, and actually not contradicting that in 8, so we kept it.

References

[1] M. H. Ali and B. Imperiali. Protein oligomerization: how and why. *Bioorganic and*

*medicinal chemistry*, 13(17):5013–5020, 2005.

[2] B. D. Bennett, E. H. Kimball, M. Gao, R. Osterhout, S. J. Van Dien, and J. D. Rabinowitz.

Absolute metabolite concentrations and implied enzyme active site occupancy

in *Escherichia coli*. *Nature chemical biology*, 5(8):593–599, 2009.

[3] M. Doebeli, E. C. Jaque, and Y. Ispolatov. Boom-bust population dynamics increase

diversity in evolving competitive communities. *Communications biology*, 4(1):1–8,

2021.

[4] C. Giersch. Determining elasticities from multiple measurements of flux rates and

metabolite concentrations: application of the multiple modulation method to a reconstituted pathway. *European journal of biochemistry*, 227(1-2):194–201, 1995.

[5] I. Jarmoskaite, I. AlSadhan, P. P. Vaidyanathan, and D. Herschlag. How to measure and evaluate binding affinities. *eLife*, 9:e57264, 2020.

[6] A. J. Loder, Y. Han, A. B. Hawkins, H. Lian, G. L. Lipscomb, G. J. Schut, M. W. Keller, M. W. Adams, and R. M. Kelly. Reaction kinetic analysis of the 3-hydroxypropionate/4-ydroxybutyrate co2 fixation cycle in extremely thermoacidophilic archaea. *Metabolic engineering*, 38:446–463, 2016.

[7] R. Milo, P. Jorgensen, U. Moran, G. Weber, and M. Springer. Bionumbersthe database

of key numbers in molecular and cell biology. *Nucleic acids research*, 38(suppl 1):

D750–D753, 2010.

[8] A. Posfai, T. Taillefumier, and N. S. Wingreen. Metabolic trade-offs promote diversity

in a model ecosystem. *Phys. Rev. Lett.*, 118:028103, Jan 2017.

[9] J. M. Rohwer, N. D. Meadow, S. Roseman, H. V. Westerhoff, and P. W. Postma. Understanding glucose transport by the bacterial phosphoenolpyruvate: glycose phosphotransferase system on the basis of kinetic measurements in vitro. *Journal of Biological Chemistry*, 275(45):34909–34921, 2000.

[10] R. Rutkis, U. Kalnenieks, E. Stalidzans, and D. A. Fell. Kinetic modelling of the

zymomonasmobilisentner–doudoroff pathway: insights into control and functionality.

*Microbiology*, 159(Pt 12):2674–2689, 2013.

[11] E. Saavedra, R. Encalada, E. Pineda, R. Jasso-Ch´avez, and R. Moreno-S´anchez. Glycolysis in entamoeba histolytica: biochemical characterization of recombinant glycolytic enzymes and flux control analysis. *The FEBS journal*, 272(7):1767–1783, 2005.

[12] M. A. Savageau. Biochemical systems analysis: I. some mathematical properties of the rate law for the component enzymatic reactions. *Journal of theoretical biology*, 25(3): 365–369, 1969.

[13] A. Schmidt, K. Kochanowski, S. Vedelaar, E. Ahrn´e, B. Volkmer, L. Callipo,

K. Knoops, M. Bauer, R. Aebersold, and M. Heinemann. The quantitative and

condition-dependent *Escherichia coli* proteome. *Nature biotechnology*, 34(1):104–110,

2016.

[14] Y. Sun and S. Qian. Flux control analysis for biphenyl metabolism by rhodococcus

pyridinovorans r04. *Biotechnology letters*, 24(18):1525–1529, 2002.

[15] T. W. Traut. Dissociation of enzyme oligomers: a mechanism for allosteric regulation.

*Critical reviews in biochemistry and molecular biology*, 29(2):125–163, 1994.

[16] J. Van Der Vlag, R. Van’t Hof, K. Van Dam, and P. W. Postma. Control of glucose

metabolism by the enzymes of the glucose phosphotransferase system in *Salmonella*

typhimurium. *European journal of biochemistry*, 230(1):170–182, 1995.